# Promises and Pitfalls of Threshold-based Auto-labeling

**Harit Vishwakarma**
hvishwakarma@cs.wisc.edu
University of Wisconsin-Madison

**Heguang Lin**
hglin@seas.upenn.edu
University of Pennsylvania

**Frederic Sala**
fredsala@cs.wisc.edu
University of Wisconsin-Madison

**Ramya Korlakai Vinayak**
ramya@ece.wisc.edu
University of Wisconsin-Madison

## Abstract

Creating large-scale high-quality labeled datasets is a major bottleneck in supervised machine learning workflows. Threshold-based auto-labeling (TBAL), where validation data obtained from humans is used to find a confidence threshold above which the data is machine-labeled, reduces reliance on manual annotation. TBAL is emerging as a widely-used solution in practice. Given the long shelf-life and diverse usage of the resulting datasets, understanding when the data obtained by such auto-labeling systems can be relied on is crucial. This is the first work to analyze TBAL systems and derive sample complexity bounds on the amount of human-labeled validation data required for guaranteeing the quality of machine-labeled data. Our results provide two crucial insights. First, reasonable chunks of unlabeled data can be automatically and accurately labeled by seemingly bad models. Second, a hidden downside of TBAL systems is potentially prohibitive validation data usage. Together, these insights describe the promise and pitfalls of using such systems. We validate our theoretical guarantees with extensive experiments on synthetic and real datasets[1].

## 1 Introduction

Machine learning (ML) models with millions or even billions of parameters are used to obtain state-of-the-art performance in many applications, e.g., object identification [53], machine translation [63], and fraud detection [71]. Such large-scale models require training on large-scale labeled datasets. As an outcome, the typical supervised ML workflow begins with the construction of a large-scale high quality dataset. Datasets with up to millions of labeled data points have played a pivotal role in the advancement of computer vision. However, collecting labeled data is an expensive and time consuming process. A common approach is to rely on the services of crowdsourcing platforms such as Amazon Mechanical Turk (AMT) to get groundtruth labels.

Even with crowdsourcing, obtaining labels for the entire dataset is expensive. To reduce costs, data labeling systems that partially rely on using a model's predictions as labels have been developed. Such systems date back to teacher-less training [21]. Modern examples include Amazon Sagemaker Ground Truth [58] and others [61, 55, 2, 64]. These approaches can be broadly termed *auto-labeling*.

Auto-labeling systems aim to label unlabeled data using predictions from ML models that are often trained on small amounts of human labeled data which can produce incorrect labels. The shelf life of datasets is longer than those of models, e.g., ImageNet continues to be a benchmark for many computer vision tasks [14] fifteen years after its initial development. As a result, to reliably train new models on auto-labeled datasets and deploy them, we need a thorough understanding of how reliable the datasets output by these auto-labeling systems are. Unfortunately, many widely

---

[1]https://github.com/harit7/TBAL

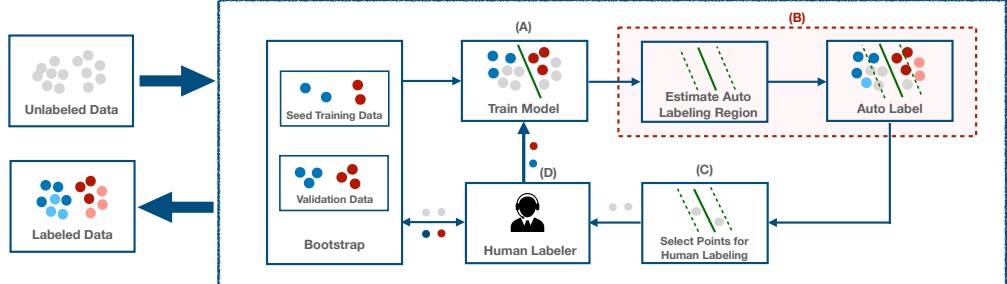

Figure 1: High-level workflow threshold-based auto-labeling (TBAL). Box (B) shows the key component estimating the auto-labeling region using validation data and auto-labeling points in it.

used commercial auto-labeling systems [58, 55] are largely opaque with limited public information on their functionality. It is therefore unclear whether the quality of the datasets obtained can be trusted. To address this, we study the high level workflow of a popular *threshold-based* auto-labeling (TBAL) system (see Figure 1). We emphasize that our goal is to understand such systems and their performance—not to promote them as a superior alternative to other approaches. Our goal is:

> **Goal.** Develop a fundamental understanding of TBAL systems. This is crucial since there is a lack of theoretical understanding of the reliability of these systems despite their wide adoption.

The TBAL systems we study (Figure 1) work iteratively. At a high level, in each iteration, the system trains a model on currently available human labeled data and decides to label certain parts of unlabeled data using the trained model by finding high-accuracy regions using validation data. It then collects human labels on a small portion of unlabeled data that is deemed helpful for training the current model in the next iteration. The validation data is created by sampling i.i.d. points from the unlabeled pool and querying human labels for them. In addition to training data, the validation data is a major driver of the cost and accuracy of auto-labeling and will be a key component in our study.

**Our Contributions.** We study TBAL systems (Figure 1) and make the following contributions:

- Provide the **first theoretical characterization of TBAL systems**, developing tradeoffs between the quantity of manually labeled data and the quantity and quality of auto-labeled data (Section 3).

- Empirical results validating our theoretical understanding on real and synthetic data (Section 4).

Our results reveal **two important insights**. Promisingly, even poor quality models are capable of reliably labeling at least some data when we have access to sufficient validation data and a good confidence function that can accurately quantify the confidence of a given model on any data point. On the downside, in certain scenarios, the quantity of the validation data required to reach a certain quantity and quality of auto-labeled data can be high.

## 2 Threshold-Based Auto-Labeling Algorithm

We begin with the problem setup and describe the TBAL algorithm that is inspired by the commercial systems [58]. Then we provide experiments and theoretical analysis shedding light on the pros and cons of TBAL. We emphasize that TBAL is not our proposal and our goal is to *understand* the effectiveness of such an auto-labeling system.

### 2.1 Problem Setup

**Notation.** Let the instance and label spaces be $\mathcal{X}$ and $\mathcal{Y} = \{1, \ldots, k\}$. We assume that there is some *deterministic* but unknown function $f^* : \mathcal{X} \mapsto \mathcal{Y}$ that assigns true label $y = f^*(\mathbf{x})$ to any $\mathbf{x} \in \mathcal{X}$. We also assume that there is a *noiseless oracle* $\mathcal{O}$ that can provide the true label $y \in \mathcal{Y}$ for any given $\mathbf{x} \in \mathcal{X}$. Let $X_{pool} \subseteq \mathcal{X}$ denote a sufficiently large pool of unlabeled data to be labeled.

The goal of an auto-labeling algorithm is to produce accurate labels $\tilde{y}_i \in \mathcal{Y}$ for points $\mathbf{x}_i \in X_{pool}$ while minimizing the number of queries to the oracle. Let $[m] := \{1, 2, \ldots, m\}$, $A \subseteq [N]$ be the set of indices of auto-labeled points, and $X_{pool}(A)$ be these points. The *auto-labeling error* $\widehat{\mathcal{E}}(X_{pool}(A))$

and the *coverage* $\widehat{\mathcal{P}}(X_{pool}(A))$ are defined as

$$\widehat{\mathcal{E}}(X_{pool}(A)) := \frac{1}{N_a} \sum_{i \in A} \mathbb{1}(\tilde{y}_i \neq f^*(\mathbf{x}_i)) \text{ and } \widehat{\mathcal{P}}(X_{pool}(A)) := \frac{|A|}{N} = \frac{N_a}{N}, \quad (1)$$

where $N_a$ denotes the size of auto-labeled set $A$. TBAL algorithm aims to auto-label the dataset so that $\widehat{\mathcal{E}}(X_{pool}(A)) \leq \epsilon_a$ while maximizing coverage $\widehat{\mathcal{P}}(X_{pool}(A))$ for any given $\epsilon_a \in (0,1)$.

**Hypothesis Class and Confidence Function.** A TBAL algorithm is given a fixed *hypothesis space* $\mathcal{H}$ and a *confidence function* $g : \mathcal{H} \times \mathcal{X} \mapsto T \subseteq \mathbb{R}^+$ that quantifies the confidence of $h \in \mathcal{H}$ on any data point $\mathbf{x} \in \mathcal{X}$. Confidence functions include prediction probabilities and margin scores. For example, when $\mathcal{H}$ is a set of unit-norm homogeneous linear classifiers, i.e. $h_{\mathbf{w}}(\mathbf{x}) = \text{sign}(\mathbf{w}^T\mathbf{x})$ with $\mathbf{w} \in \{\mathbf{w} \in \mathbb{R}^d : ||\mathbf{w}||_2 = 1\}$, a reasonable confidence function is $g(h_{\mathbf{w}}, \mathbf{x}) = |\mathbf{w}^T\mathbf{x}|$.

Note that the target $f^*$ might not be in the hypothesis space $\mathcal{H}$. Our analysis (Section 3) shows that the TBAL algorithm can work well, i.e., accurately label a reasonable fraction of unlabeled data with simpler hypothesis classes that do not contain the target hypothesis $f^*$. We illustrate this with a simple example in Section 2.3 and Figure 2. Note as well that the features $\mathbf{x}$ could be raw features or representations from self-supervised techniques, pre-trained models, etc. We analyze TBAL in settings (i) with no assumptions on the features and (ii) when the features are linearly separable.

## 2.2 Description of the algorithm

The TBAL algorithm is given in Algorithm 1. It starts with an unlabeled pool $X_{pool}$ and an auto-labeling error threshold $\epsilon$. For ease of exposition, the algorithm is given the labeled validation set $D_{val}$ of size $N_v$ separately. In practice, it is created by selecting points

---

**Algorithm 1** Threshold-based Auto-Labeling (TBAL)

**Input:** Unlabeled pool $X_{pool}$, auto labeling error threshold $\epsilon_a$, seed data size $n_s$, batch size for active query $n_b$, labeled validation data pool $D_{val}$.

**Output:** $D_{out} = \{(\mathbf{x}_i, \tilde{y}_i) : \forall \mathbf{x}_i \in X_{pool}\}$

1: $X_u^{(1)} = X_{pool}; D_{val}^{(1)} = D_{val}$
2: $D_{query}^{(1)} = \texttt{randomly\_query\_batch}(X_u^{(1)}, n_s)$
3: Remove queried points from $X_u^{(1)}$
4: $D_{train}^{(0)} = \phi; i = 1; D_{out} = D_{out}^{(1)} = D_{query}^{(1)}$
5: **while** $X_u^{(i)} \neq \phi$ **do**
6: $\quad D_{train}^{(i)} = D_{train}^{(i-1)} \cup D_{query}^{(i)}$
7: $\quad \hat{h}_i = \texttt{empirical\_risk\_min}(\mathcal{H}, D_{train}^{(i)})$
8: $\quad \hat{t}_i = \text{Estimate Threshold}(X_u^{(i)}, D_{val}^{(i)}, \epsilon_a, \hat{h}_i, n_0)$
9: $\quad D_{auto}^{(i)} = \{(\mathbf{x}, \hat{h}_i(\mathbf{x})) : \mathbf{x} \in X_u^{(i)}, g(\hat{h}_i, \mathbf{x}) \geq \hat{t}_i\}$
10: $\quad X_u^{(i+1)} = \{\mathbf{x} : \mathbf{x} \in X_u^{(i)}, g(\hat{h}_i, \mathbf{x}) < \hat{t}_i\}$
11: $\quad D_{val}^{(i+1)} = \{(\mathbf{x}, y) : (\mathbf{x}, y) \in D_{val}^{(i)}, g(\hat{h}_i, \mathbf{x}) < \hat{t}_i\}$
12: $\quad D_{query}^{(i+1)} = \texttt{active\_query\_batch}(\hat{h}_i, X_u^{(i+1)}, n_b)$
13: $\quad$ Remove queried points from $X_u^{(i+1)}$
14: $\quad D_{out} = D_{out} \cup D_{auto}^{(i)} \cup D_{query}^{(i+1)}$
15: $\quad i = i + 1$
16: **end while**

---

**Algorithm 2** Estimate Threshold

**Input:** $X_u^{(i)}, D_{val}^{(i)}, \epsilon_a, \hat{h}_i, n_0$
**Output:** Threshold $\hat{t}_i$

1: $T_i = \{g(\hat{h}_i, \mathbf{x}) : (\mathbf{x}, y) \in D_{val}^{(i)}\}$
2: **for** $t \in T_i$ **do**
3: $\quad N_t = |\{\mathbf{x} \in X_v^{(i)} : g(\hat{h}_i, \mathbf{x}) \geq t\}|$
4: **end for**
5: $\tilde{T}_i = \{t : t \in T_i, N_t \geq n_0\} \cup \{\infty\}$
6: $\hat{t}_i = \min\{t \in \tilde{T}_i : \widehat{\mathcal{E}}_a(\hat{h}_i, t|X_v^{(i)}) + C_1\hat{\sigma}_i \leq \epsilon_a\}$

---

at random from $X_{pool}$. The algorithm starts with an initial batch of $n_s$ random data points and obtains oracle labels for these. The algorithm works in an iterative manner using the following steps.

1. Oracle labeled data obtained in each iteration $i$ is added to the training pool $D_{train}^{(i)}$. It is used to train a model $\hat{h}_i$ by performing empirical risk minimization (ERM).

2. Find the region where $\hat{h}_i$ can auto-label accurately. The algorithm estimates a threshold $\hat{t}_i$ on the confidence score above which it can auto-label with the desired auto-labeling accuracy on the validation data (see Algorithm 2). Thresholds that have too little validation data are discarded, since their estimates are unreliable. The minimum threshold is found such that the sum of the estimated error $\widehat{\mathcal{E}}_a(\hat{h}_i, t|X_v^{(i)})$ (see eq. (2)) and an upper confidence bound using the standard deviation of the estimated error, is at most the given auto-labeling error threshold.

3. Auto-label the points in the pool, $X_u^{(i)}$, which have confidence $g(\hat{h}_i, \mathbf{x}) > \hat{t}_i$. These are added to the set $D_{out}$ and removed from the unlabeled pool. The validation points that fall in the auto-labeled region are also removed from the validation set so that in the next round the validation

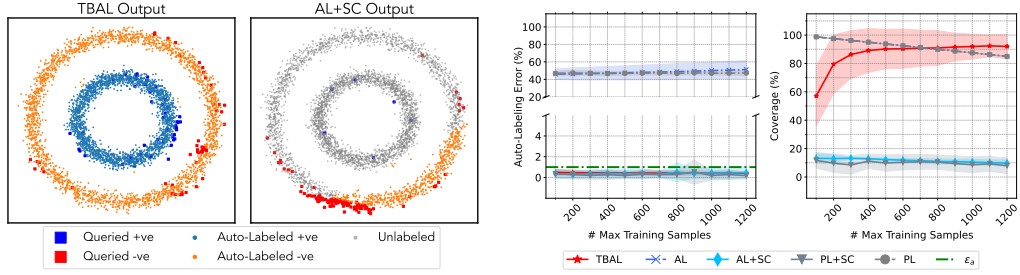

(a) TBAL and AL+SC on Circles dataset      (b) Auto labeling performance of various methods

Figure 2: Comparison of TBAL, active learning (AL) followed by selective classification (AL+SC) and passive learning (PL) followed b selective classification (PL+SC) on the Circles dataset (Sec. 2.3) using linear classifiers and confidence functions. (a) Samples auto-labeled, queried, and left unlabeled. (b) The auto-labeling error and coverage achieved by the algorithms. (50 trials.)

set and the unlabeled pool are from the same region and the same distribution. Removing the auto-labeled points from $X_{\text{pool}}$ is a crucial step in the TBAL algorithm that enables it to focus only on the remaining unlabeled regions in the next iteration.

4. If there are points left in $X_{\text{pool}}$, the algorithm selects points using some active querying strategy [57], obtains human labels for them, and adds them to the training pool. Note that the auto-labeled data is not added into the training set.

This process continues until there are no data points left to be labeled. The algorithm then outputs the labeled dataset, which is a mixture of human- and machine-labeled points.

## 2.3 Comparison between Auto-Labeling, Active Learning and Selective Classification

What is the difference between TBAL and methods such as active learning and selective classification?

*Active learning.* The goal of active learning [57] (AL) is to find the best model in hypothesis class $\mathcal{H}$ by training with less labeled data compared to passive learning. This is usually achieved by obtaining labels for the most informative points. Note that the *end goal is to output a model* from the function class whose predictions on new data as good as the best model in the function class could.

*Selective Classification.* The goal of selective classification (SC) [17] is to find the best combination of the hypothesis and selection functions to minimize error and maximize coverage of selection regions.

*Auto-Labeling.* The output of an auto-labeling procedure is a labeled dataset (not a model). When the hypothesis class is of lower complexity, it is often not possible to find a good classifier. The goal of an auto-labeling system is to label as much of the unlabeled data as accurately as possible with a given function class and with limited labeled data from humans.

**Is active learning alone enough to auto-label data?** AL has been found to be effective in reducing the number of labels needed to learn versus passive learning, particularly in low-noise cases [28]. Doing auto-labeling using AL followed by SC may be effective in such settings. However, in real-world scenarios, noise levels may be higher and the hypothesis class could be misspecified. In these instances, using the model learned through active learning to automatically label all data may result in a high number of errors.

We illustrate this difference between AL, SC, and auto-labeling through an example. Suppose the data consists of two concentric circles, one for each class, with the same number of points per class (Figure 2(a)). This setting is not linearly separable. We run TBAL, AL, and AL followed by SC with an error tolerance of $\epsilon_a = 1\%$ and linear classifiers and confidence functions. The results are shown in Figure 2. Note that the multiple optimal linear classifiers will all incur an error of $50\%$. AL algorithms can only output models that make at least $50\%$ error. If we naively use the output model for auto-labeling, we can obtain near full coverage but incur around $50\%$ auto-labeling error. If we use the model output by AL with threshold-based SC, labeling error is reduced. However, it can only label $\approx 25\%$ of the unlabeled data. In contrast, TBAL can label almost all of the data accurately (close to $100\%$ coverage) with less than $1\%$ auto-labeling error.

# 3 Theoretical Analysis

The performance of the TBAL (Algorithm 1) depends on many factors including the hypothesis class, the accuracy of the confidence function, the data sampling strategy, and the size of the training and validation data. In particular, the amount of validation data plays a critical role in determining the accuracy of the confidence function, which in turn affects the accuracy and coverage.

We derive bounds on the auto-labeling error and the coverage for Algorithm 1 in terms of the size of the validation data, the number of auto-labeled points $N_a^{(k)}$, and the Rademacher complexity of the extended hypothesis class $\mathcal{H}^{T,g}$ induced by the confidence function $g$. Our first result, Theorem (3.2), applies to general settings and makes no assumptions on the particular form of the hypothesis class, the data distribution, and the confidence function. We then instantiate and specialize the results for a specific setting in Section 3.1. We introduce some notation to aid in stating our results,

**Definition 3.1.** *(Hypothesis Class with Abstain) The function $g$, set $T$ and $\mathcal{H}$ induce an extended hypothesis class $\mathcal{H}^{T,g} := \mathcal{H} \times T$. Any function $(h,t) \in \mathcal{H}^{T,g}$ is defined as $(h,t)(\mathbf{x}) = h(\mathbf{x})$ if $g(h,\mathbf{x}) \geq t$ and $\perp$ otherwise. Here $(h,t)(\mathbf{x}) = \perp$ means $(h,t)$ abstains in classifying the point $\mathbf{x}$.*

**Error Definitions.** Let $\mathcal{S} \subseteq \mathcal{X}$ denote a non-empty sub-region of $\mathcal{X}$ and $S \subseteq \mathcal{S}$ be a finite set of i.i.d. samples from $\mathcal{S}$. The subset $\mathcal{S}(h,t) \subseteq \mathcal{S}$ denotes the regions where $(h,t)$ does not abstain i.e. $\mathcal{S}(h,t) := \{\mathbf{x} \in \mathcal{S} : (h,t)(\mathbf{x}) \neq \perp\}$, and the conditional probability mass associated with it is $\mathbb{P}(h,t|\mathcal{S}) := \mathbb{P}(\mathcal{S}(h,t)|\mathcal{S})$, and its empirical counterpart $\widehat{\mathbb{P}}(h,t|S) := |S(h,t)|/|S|$. We use $\mathbb{P}(\mathcal{S})$ to denote the probability mass of set $\mathcal{S}$ and $\mathbb{P}(\mathcal{S}'|\mathcal{S})$ for the conditional probability of subset $\mathcal{S}' \subseteq \mathcal{S}$ given $\mathcal{S}$. The population level and empirical auto-labeling errors are defined as follows:

$$\mathcal{E}_a(h,t|\mathcal{S}) := \mathbb{E}_{\mathbf{x}|\mathcal{S}}[\ell_{0-1}^{\perp}(h,t,\mathbf{x},y)]/\mathbb{P}(h,t|\mathcal{S}),$$

$$\widehat{\mathcal{E}}_a(h,t|S) := \left(\sum_{\mathbf{x}_i \in S(h,t)} \ell_{0-1}^{\perp}(h,t,\mathbf{x}_i,y_i)\right)/|S(h,t)| \tag{2}$$

Here $\ell_{0-1}^{\perp}(h,t,\mathbf{x},y) := \ell_{0-1}(h,\mathbf{x},y) \cdot \ell_{\perp}(h,t,\mathbf{x})$ with $\ell_{0-1}(h,\mathbf{x},y) := \mathbb{1}(h(\mathbf{x}) \neq y)$, and $\ell_{\perp}(h,t,\mathbf{x}) := \mathbb{1}(g(h,\mathbf{x}) \geq t)$.

**Rademacher Complexity.** The Rademacher complexities for the function classes induced by the $\mathcal{H}, T, g$ and the loss functions are defined as $\mathfrak{R}_n(\mathcal{H}^{T,g}) := \mathfrak{R}_n(\mathcal{H}, \ell_{0-1}) + \mathfrak{R}_n(\mathcal{H}^{T,g}, \ell_{\perp})$. Let $\hat{h}_i$ and $\hat{t}_i$ be the ERM solution and the auto-labeling threshold at epoch $i$. Let $p_0 \in (0,1)$ be a constant such that $\mathbb{P}(\hat{h}_i, \hat{t}_i|\mathcal{X}^{(i)}) \geq p_0$ for all $i$. Let $X_v^{(i)}$ denote the validation set, and $n_v^{(i)}$ and $n_a^{(i)}$ the number of validation and auto-labeled points at epoch $i$. Let $\widehat{\mathcal{E}}_a(\hat{h}_i, \hat{t}_i|X_v^{(i)})$ be the empirical conditional risk of $\hat{h}_i$ in the region where $g(\hat{h}_i, \mathbf{x}) \geq \hat{t}_i$ evaluated on the validation data $X_v^{(i)}$.

We provide the following guarantees on the auto-labeling error and the coverage achieved by TBAL.

**Theorem 3.2.** *(Overall Auto-Labeling Error and Coverage) Let $k$ denote the number of rounds of the TBAL Algorithm 1. Let $n_v^{(i)}, n_a^{(i)}$ denote the number of validation and auto-labeled points at epoch $i$ and $n^{(i)} = |X^{(i)}|$. Let $X_{pool}(A_k)$ be the set of auto-labeled points at the end of round $k$. $N_a^{(k)} = \sum_{i=1}^k n_a^{(i)}$ be the total number of auto-labeled points. Then, for any $\delta \in (0,1)$, with probability at least $1 - \delta$,*

$$\widehat{\mathcal{E}}(X_{pool}(A_k)) \leq \sum_{i=1}^k \frac{n_a^{(i)}}{N_a^{(k)}}\Big(\underbrace{\widehat{\mathcal{E}}_a(\hat{h}_i, \hat{t}_i|X_v^{(i)})}_{(a)} + \underbrace{\frac{4}{p_0}\big(\mathfrak{R}_{n_v^{(i)}}(\mathcal{H}^{T,g}) + \frac{2}{p_0}\sqrt{\frac{1}{n_v^{(i)}}\log(\frac{8k}{\delta})}\big)}_{(b)}\Big)$$

$$+ \underbrace{\frac{4}{p_0}\Big(\sum_{i=1}^k \frac{n_a^{(i)}}{N_a^{(k)}}\mathfrak{R}_{n_a^{(i)}}(\mathcal{H}^{T,g}) + \sqrt{\frac{k}{N_a^{(k)}}\log(\frac{8k}{\delta})}\Big)}_{(c)}, \quad and$$

$$\widehat{\mathcal{P}}(X_{pool}(A_k)) \geq \sum_{i=1}^k \mathbb{P}(\mathcal{X}^{(i)}(\hat{h}_i, \hat{t}_i)) - 2\mathfrak{R}_{n^{(i)}}(\mathcal{H}^{T,g}) - \sqrt{\frac{2k^2}{N}\log\left(\frac{8k}{\delta}\right)}.$$

**Discussion.** We interpret this result, starting with the auto-labeling error term $\widehat{\mathcal{E}}(X_{pool}(A_k))$. The term (a) $\widehat{\mathcal{E}}_a(\hat{h}_i, \hat{t}_i|X_v^{(i)})$ is the empirical conditional error in the auto-labeled region computed on the validation data in $i$-th round, which is at most $\epsilon_a$. Thus, summing term (a) over all the rounds is at most $\epsilon_a$. The term (b) provides an upper bound on the excess error over the empirical estimate term

(a) as a function of the Rademacher complexity of $\mathcal{H}^{T,g}$ and the validation data used in each round. The last term (c) captures the variance in the overall estimate as a function of the total number of auto-labeled points and the Rademacher complexity of $\mathcal{H}^{T,g}$. If we let $n_v^{(i)} \geq n_v$ i.e. the minimum validation points ensured in each round, then we can see the second term is $\mathcal{O}(\mathfrak{R}_{n_v}(\mathcal{H}^{T,g}))$ and the third term is $\mathcal{O}(\sqrt{1/n_v})$.

Therefore, validation data of size $\mathcal{O}\left(1/\epsilon_a^2\right)$ in each round is sufficient to get a $\mathcal{O}(\epsilon_a)$ bound on the excess auto-labeling error. The terms with Rademacher complexities suggest that it is better to use a hypothesis class and confidence function such that the induced hypothesis class has low Rademacher complexity. While such a hypothesis class might not be rich enough to include the target function, it would still be helpful for efficient and accurate auto-labeling of the dataset which can then be used for training richer models in the downstream task. The coverage term provides a lower bound on the empirical coverage $\widehat{\mathcal{P}}(X_{pool}(A_k))$ in terms of the true coverage of the sequence of estimated hypotheses $\hat{h}_i$ and threshold $\hat{t}_i$.

We note that the size of the validation data needed to guarantee the auto-labeling error in each round by Algorithm 1 is optimal up to $\log$ factors. This follows by applying a result on the tail probability of the sum of independent random variables due to Feller [20]:

**Lemma 3.3.** *Let $c_1, c_2$ and $\sigma > 0$. Let $\mathbf{x}_i \in X$ be a set of $n$ i.i.d. points from $\mathcal{X}$ with corresponding true labels $y_i$. Given $(h, t) \in \mathcal{H}^{T,g}$, let $\mathbb{E}\left[\left(\ell_{0-1}^{\perp}(h, t, \mathbf{x}_i, y_i) - \mathcal{E}(h, t|\mathcal{X})\right)^2\right] = \sigma_i^2 > \sigma^2$ for every $\mathbf{x}_i$ for $\sigma_i > 0$ and let $\sum_i^n \sigma_i^2 \geq c_1$ then for every $\epsilon \in [0, (\sum_{i=1}^n \sigma_i^2)/\sqrt{c_1}]$ with $n_v < 12\sigma^2 \log(4c_2)/\epsilon^2$, the following holds w.p. at least $1/4$, $\mathcal{E}_a(h, t|\mathcal{X}) > \widehat{\mathcal{E}}_a(h, t|X) + \epsilon$.*

Therefore, if a sufficiently large validation set is not used in each round, there is a constant probability of erroneously deciding on a threshold for auto-labeling. Such a requirement on validation data also applies to active learning if we seek to validate the output model. Bypassing this requirement demands the use of approaches that are different from threshold-based auto-labeling and traditional validation techniques. We note the possibility of using recently proposed *active testing* techniques [33], a nascent approach to reducing validation data usage.

## 3.1   Linear Classifier Setting

Next, we consider a simple setting where active learning is known to be optimal to see if TBAL can offer similar performance guarantees. To do so, we instantiate results from 3.2 to homogeneous linear separators under the uniform distribution in the realizable setting. Let $P_{\mathbf{x}}$ be supported on the unit ball in $\mathbb{R}^d$, $\mathcal{X} = \{\mathbf{x} \in \mathbb{R}^d : ||\mathbf{x}|| \leq 1\}$. Let $\mathcal{W} = \{\mathbf{w} \in \mathbb{R}^d : ||\mathbf{w}||_2 = 1\} = \mathbb{S}_d$, $\mathcal{H} = \{\mathbf{x} \mapsto \text{sign}(\langle \mathbf{w}, \mathbf{x} \rangle) \forall \mathbf{w} \in \mathcal{W}\}$, the score function be given by $g(h, \mathbf{x}) = g(\mathbf{w}, \mathbf{x}) = |\langle \mathbf{w}, \mathbf{x} \rangle|$, and set $T = [0, 1]$. For simplicity, we will use $\mathcal{W}$ in place of $\mathcal{H}$.

**Corollary 3.4.** *(Overall Auto-Labeling Error and Coverage) Let $\hat{\mathbf{w}}_i, \hat{t}_i$ be the ERM solution and the auto-labeling threshold respectively at epoch $i$. Let $n_v^{(i)}, n_a^{(i)}$ denote the number of validation and auto-labeled points at epoch $i$. Let the TBAL algorithm run for $k$-epochs. Then, for any $\delta \in (0, 1)$, w.p. at least $1 - \delta$,*

$$\widehat{\mathcal{E}}(X_{pool}(A_k)) \leq \sum_{i=1}^k \frac{n_a^{(i)}}{N_a^{(k)}} \left( \underbrace{\widehat{\mathcal{E}}_a(\hat{\mathbf{w}}_i, \hat{t}_i | X_v^{(i)})}_{(a)} + \underbrace{\frac{4}{P_0} \sqrt{\frac{2}{n_v^{(i)}} \left( 2d \log\left(\frac{e n_v^{(i)}}{d}\right) + \log\left(\frac{8k}{\delta}\right) \right)}}_{(b)} \right)$$

$$+ \underbrace{\frac{4}{P_0} \left( \sqrt{\frac{2k}{N_a^{(k)}} \left( 2d \log\left(\frac{e N_a^{(k)}}{d}\right) + \log\left(\frac{8k}{\delta}\right) \right)} \right)}_{c}, \quad and$$

$$\widehat{\mathcal{P}}(X_{pool}(A_k)) \geq 1 - \min_i \hat{t}_i \sqrt{4d/\pi} - 2k \sqrt{\frac{2}{N} \left( 2d \log\left(\frac{e N}{d}\right) + \log\left(\frac{8k}{\delta}\right) \right)}.$$

These results imply that by ensuring the sum of the empirical validation error term (a) and the upper confidence interval to be less than $\epsilon_a$ in each round of the algorithm we can ensure that the overall auto-labeling error remains below $\epsilon_a$. Furthermore, by applying standard VC theory to the first round, we obtain that $\hat{t}_1 \leq 1/2$. Therefore, right after the first round, we are guaranteed to label at least half of the unlabeled pool. We empirically observe that TBAL has coverage at par with active learning while respecting the auto-labeling error constraint (See Figure 4(a)).

**Tightness of the Bounds.** We study this in the setting of the Unit-Ball experiment. The upper bound on excess risk in this setting is given in Corollary 3.4 which is an instantiation of our general results to this specific setting. We consider a simplified form of the upper bound by ignoring the constants to get a sense of the rate in terms of the validation data size. We compute this simplified upper bound for different amounts of validation data. We compare these with the maximum auto-labeling error observed over 25 runs of auto-labeling in the Unit-Ball setting with different random seeds

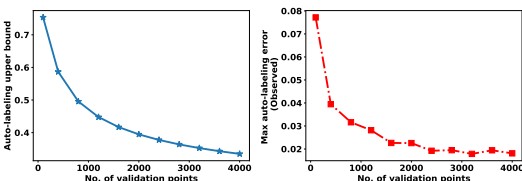

Figure 3: **Left:** Simplified upper bound from Corollary 3.4 (ignoring constants) on excess auto-labeling error for the Unit-Ball setting i.e. homogeneous linear classifier with $d = 30$. **Right:** The worst observed auto-labeling error over 25 trials in the Unit-Ball experiment.

for each validation data size. The results are in Figure 3. As expected, we see that the worst-case error rate follows a similar rate as our upper bound but the upper bound is conservative. Next, we explain why this is the case.

Our upper bound is slightly conservative, as it is based on a uniform bound over all hypotheses in a given hypothesis class. Since the individual hypotheses whose excess auto-labeling error we need to bound are not known a priori we need to derive a bound on the number of validation samples using which we can guarantee that the excess auto-labeling error of any hypothesis (model) is small with high probability. Note that this is a conservative (worst-case) analysis to get an upper bound on the validation sample complexity. The upper bound has two parts: a) the Rademacher complexity of the hypothesis class and b) a term with the number of validation samples. We note that on the validation samples, it matches lower bounds order-wise (see Lemma 3.3). This is the first analysis to provide these bounds based on uniform convergence without making any assumptions about the data distributions or hypothesis class. We provide further discussion on the role of Rademacher complexity in the Appendix C.1.

## 4 Experiments

We study the effectiveness of TBAL on synthetic and real datasets. We validate our theoretical results and aim to understand the amount of labeled validation and training data required to achieve a certain auto-labeling error and coverage. We also seek to understand whether our findings apply to real data—where labels may be noisy—along with how TBAL performs compared to common baselines.

**Baselines.** We compare TBAL to the following methods:,

a) *Passive Learning (PL)* queries a subset of the points randomly to train a model from a given model class and then uses it to predict the labels for the remaining unlabeled pool.

b) *Active Learning (AL)* (using margin-random query strategy, described below) trains a model from a model class and uses it to predict the labels for the remaining unlabeled pool.

c) *Passive Labeling + Selective Classification (PL+SC)* first performs passive learning to train a model from a given model class. Then it performs auto-labeling on the unlabeled data using threshold-based selective classification with the model output by passive learning. Only those unlabeled points that are deemed as fit to be labeled by the selection function are auto-labeled.

d) *Active Learning + Selective Classification (AL+SC)* first performs active learning (using margin-random query strategy) to train a model from a given model class. It then performs auto-labeling using threshold-based selective classification with the model output by AL.

For selective classification in the above methods, we use Algorithm 2 to estimate the threshold and use it to perform auto-labeling. In experiments, we modify Algorithm 2 slightly—instead of estimating a single threshold for all classes, we estimate thresholds for each class separately.

**Active Querying Strategy.** We use the *margin-random* query strategy for querying the next batch of training data. In this strategy, the algorithm first sorts the points based on the margin (uncertainty) score and then selects the top $Cn_b$ ($C > 1$) points from which $n_b$ points are picked at random. This is a simple and computationally efficient method that balances the exploration and exploitation trade-off. We note that other active-querying strategies exist; we use margin-random as our standard querying strategy to keep the focus on comparing auto-labeling—not active learning approaches.

| $N_v$ | Error (%) | | Coverage (%) | |
|---|---|---|---|---|
| | TBAL | AL+SC | TBAL | AL+SC |
| 100 | 3.10 $\pm$1.80 | 0.68 $\pm$0.81 | 71.43 $\pm$8.86 | 96.95 $\pm$1.01 |
| 400 | 1.97 $\pm$0.76 | 0.59 $\pm$0.18 | 93.99 $\pm$2.39 | 97.89 $\pm$0.50 |
| 800 | 1.64 $\pm$0.50 | 0.66 $\pm$0.19 | 96.26 $\pm$1.33 | 98.06 $\pm$0.53 |
| 1200 | 1.39 $\pm$0.39 | 0.67 $\pm$0.19 | 96.67 $\pm$0.84 | 98.10 $\pm$0.45 |
| 1600 | 1.33 $\pm$0.30 | 0.70 $\pm$0.19 | 97.13 $\pm$0.45 | 98.16 $\pm$0.44 |
| 2000 | 1.28 $\pm$0.34 | 0.71 $\pm$0.21 | 97.15 $\pm$0.54 | 98.20 $\pm$0.34 |

| $N_v$ | Error (%) | | Coverage (%) | |
|---|---|---|---|---|
| | TBAL | AL+SC | TBAL | AL+SC |
| 100 | 3.10 $\pm$1.80 | 0.68 $\pm$0.81 | 71.43 $\pm$8.86 | 96.95 $\pm$1.01 |
| 400 | 1.65 $\pm$0.65 | 0.32 $\pm$0.15 | 93.27 $\pm$2.50 | 96.91 $\pm$0.99 |
| 800 | 1.08 $\pm$0.47 | 0.24 $\pm$0.16 | 96.01 $\pm$1.16 | 96.31 $\pm$1.36 |
| 1200 | 0.78 $\pm$0.27 | 0.17 $\pm$0.11 | 96.82 $\pm$0.84 | 95.96 $\pm$1.40 |
| 1600 | 0.65 $\pm$0.20 | 0.13 $\pm$0.08 | 96.93 $\pm$0.57 | 95.70 $\pm$1.38 |
| 2000 | 0.54 $\pm$0.16 | 0.21 $\pm$0.11 | 97.23 $\pm$0.42 | 96.36 $\pm$1.13 |

Table 1: **Unit-Ball.** Effect of variation of validation data size ($N_v$) with and without using a UCB on error estimates. We keep training data size $N_q$ fixed at 500 and use error threshold $\epsilon_a = 1\%$. We report the mean and std. deviation over 10 runs with different random seeds. **Left**: with $C_1 = 0$. **Right**: with $C_1 = 0.25$.

| $N_v$ | Error (%) | | Coverage (%) | |
|---|---|---|---|---|
| | TBAL | AL+SC | TBAL | AL+SC |
| 200 | 4.77 $\pm$0.18 | 3.35 $\pm$0.80 | 83.14 $\pm$3.65 | 78.53 $\pm$7.05 |
| 400 | 4.57 $\pm$0.26 | 3.53 $\pm$0.73 | 90.70 $\pm$3.11 | 86.39 $\pm$5.11 |
| 600 | 4.32 $\pm$0.17 | 3.70 $\pm$0.63 | 92.96 $\pm$0.46 | 88.90 $\pm$4.83 |
| 800 | 4.66 $\pm$0.20 | 3.84 $\pm$0.70 | 92.42 $\pm$0.89 | 88.67 $\pm$3.88 |
| 1000 | 4.67 $\pm$0.16 | 3.90 $\pm$0.68 | 92.89 $\pm$0.91 | 89.79 $\pm$3.09 |

| $N_v$ | Error (%) | | Coverage (%) | |
|---|---|---|---|---|
| | TBAL | AL+SC | TBAL | AL+SC |
| 200 | 2.28 $\pm$0.21 | 3.11 $\pm$0.86 | 68.24 $\pm$6.20 | 57.77 $\pm$13.09 |
| 400 | 1.29 $\pm$0.10 | 1.98 $\pm$0.40 | 63.81 $\pm$4.86 | 63.06 $\pm$10.70 |
| 600 | 1.41 $\pm$0.20 | 1.81 $\pm$0.22 | 69.64 $\pm$3.98 | 62.92 $\pm$9.20 |
| 800 | 1.62 $\pm$0.30 | 2.04 $\pm$0.35 | 67.45 $\pm$3.72 | 63.22 $\pm$7.89 |
| 1000 | 1.64 $\pm$0.23 | 1.97 $\pm$0.26 | 70.28 $\pm$2.82 | 66.11 $\pm$8.00 |

Table 2: **IMDB.** Effect of variation of validation data size ($N_v$) with and without using a UCB on error estimates. We keep training data size $N_q$ fixed at 500 and use error threshold $\epsilon_a = 5\%$. We report the mean and std. deviation over 10 runs with different random seeds. **Left:** with $C_1 = 0$. **Right:** with $C_1 = 0.25$.

**Datasets.** We use the following synthetic and real datasets. We also provide empirical results on MNIST and another synthetic dataset in the Appendix. For each dataset, we split the data into two sufficiently large pools. One is used as $X_{pool}$ on which auto-labeling algorithms are run and the other is used as $X_{val}$ from which the algorithms subsample validation data.

a) *Unit-Ball* is a synthetic dataset of uniformly sampled points from the $d$-dimensional unit ball. The true labels are generated using a homogeneous linear separator with $\mathbf{w} = [1/\sqrt{d}, \ldots, 1/\sqrt{d}]$. We use $d = 30$ and generate $N = 20K$ samples, out of which $16K$ are in $X_{pool}$ and $4K$ are in $X_{val}$. The dataset has just two classes but there is no margin between them.

b) *Tiny-ImageNet* [1] is a subset of the larger ImageNet [14] dataset, designed for image classification tasks. It consists of *200 classes*, each with 500 training images and 50 validation and test images. With a total of $100K$ images, Tiny ImageNet provides a diverse and challenging dataset. We use pre-computed embeddings of the images using CLIP [50].

c) *IMDB Reviews* [41] is a comprehensive collection of movie reviews, consisting of $50K$ individual reviews. It is a balanced dataset of positive and negative labels. We use the standard train set of size $25K$ and split it into $X_{pool}$ and $X_{val}$ of sizes $20K$ and $5K$ respectively. We compute embeddings of reviews using a pre-trained model `bge-large-en` [70] from the Massive Text Embedding Benchmark (MTEB) [44, 19].

d) *CIFAR-10* [36] is an image dataset with 10 classes. We randomly split the standard training set into $X_{pool}$ of size $40K$ and the validation pool of size $10K$. We use the raw features for training.

**Models and Training.** For the linear models, we use SVM with the usual hinge loss and train it to loss tolerance $10^{-5}$. To train a multi-layer perceptron (MLP) on the pre-computed embeddings of IMDB and Tiny-ImageNet we use SGD with a learning rate of $0.05, 0.1$ respectively, and batch size of 64. To train the medium CNN we use SGD with a learning rate of $10^{-2}$, batch size 256, and momentum of 0.9. More details on model training are in the Appendix.

**The score function** $g$. For SVMs we use the standard implementations of [69, 46] in `sklearn` to get the prediction probabilities and use them as the score function. Neural networks use softmax output.

### 4.1 Role of Validation Data

The TBAL algorithm uses validation data to estimate the auto-labeling errors at various thresholds to determine the threshold for automatically labeling points accurately. Thus, it is crucial to have

| $N_v$ | Error (%) | | Coverage (%) | | $N_v$ | Error (%) | | Coverage (%) | |
|---|---|---|---|---|---|---|---|---|---|
| | TBAL | AL+SC | TBAL | AL+SC | | TBAL | AL+SC | TBAL | AL+SC |
| 2000 | 0.0 $_{\pm 0.0}$ | 0.0 $_{\pm 0.0}$ | 0.0 $_{\pm 0.0}$ | 0.0 $_{\pm 0.0}$ | 2000 | 0.0 $_{\pm 0.0}$ | 0.0 $_{\pm 0.0}$ | 0.0 $_{\pm 0.0}$ | 0.0 $_{\pm 0.0}$ |
| 4000 | 13.88 $_{\pm 5.42}$ | 13.31 $_{\pm 10.79}$ | 0.72 $_{\pm 0.55}$ | 0.48 $_{\pm 0.04}$ | 4000 | 10.50 $_{\pm 6.01}$ | 7.37 $_{\pm 4.57}$ | 0.47 $_{\pm 0.05}$ | 0.48 $_{\pm 0.06}$ |
| 6000 | 14.18 $_{\pm 0.76}$ | 11.52 $_{\pm 0.82}$ | 17.29 $_{\pm 0.72}$ | 8.18 $_{\pm 1.12}$ | 6000 | 10.61 $_{\pm 0.62}$ | 7.71 $_{\pm 1.03}$ | 10.16 $_{\pm 1.10}$ | 4.31 $_{\pm 1.10}$ |
| 8000 | 13.97 $_{\pm 0.14}$ | 11.31 $_{\pm 0.51}$ | 36.36 $_{\pm 1.78}$ | 23.40 $_{\pm 1.15}$ | 8000 | 9.90 $_{\pm 0.63}$ | 6.80 $_{\pm 0.77}$ | 25.84 $_{\pm 1.57}$ | 14.43 $_{\pm 2.01}$ |
| 10000 | 13.42 $_{\pm 0.29}$ | 11.14 $_{\pm 0.54}$ | 43.79 $_{\pm 0.93}$ | 33.38 $_{\pm 0.72}$ | 10000 | 8.97 $_{\pm 0.36}$ | 6.87 $_{\pm 0.48}$ | 32.19 $_{\pm 1.34}$ | 21.96 $_{\pm 1.35}$ |

Table 3: **Tiny-ImageNet.** Effect of variation of validation data size ($N_v$) with and without using a UCB on error estimates. We keep training data size $N_q$ fixed at $10k$ and use error threshold $\epsilon_a = 10\%$. We report the mean and std. deviation over 5 runs with different random seeds. **Left:** with $C_1 = 0$. **Right:** with $C_1 = 0.25$.

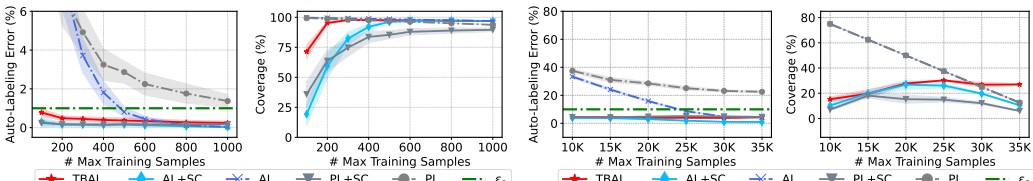

(a) Unit-Ball: varying training samples size, validation samples size=4K, $\epsilon_a = 1\%, C_1 = 0.25$.

(b) CIFAR-10: varying training samples size, validation size=10K, $\epsilon_a = 10\%, C_1 = 0.25$.

Figure 4: Results for varying $N_q$, the maximum number of samples algorithm can use for training while providing sufficient validation samples.

accurate estimates of the auto-labeling errors. Our analysis shows that to get such good estimates, large amounts of validation data are needed. In this section, we study the effect of varying the amount of validation data on auto-labeling performance.

**Setup.** We fix the maximum training data size $N_q$ and run the algorithm with different amounts of validation data. We also consider the two cases where the algorithm uses an upper confidence bound on the error estimate and where it does not. We use the Unit-Ball, IMDB, Tiny-ImageNet, and datasets for this study with $N_q = 500, 500,$ and $10K$ respectively, and the auto-labeling error thresholds $\epsilon_a = 1\%, 5\%, 10\%$, respectively. Initial seed data of size $n_s$ is 20% of $N_q$ and query batch size $n_b$ is 5% of $N_q$; $C = 2$ for both AL and TBAL for both datasets. We give the same initial seed samples of size $n_s$ to all the methods to ensure they have the same starting point.

**Results.** Tables 1,2 and 3 demonstrate the impact of validation data on the performance of TBAL and other algorithms. The auto-labeling error and coverage of TBAL and other methods are affected by the amount of validation data provided. When the validation data is insufficient, the auto-labeling error of TBAL increases. However, as more validation data is used, the auto-labeling error and coverage of TBAL improves. Providing too little $X_{val}$ can lead to incorrect estimates of the auto-labeling error, which in turn results in poor auto-labeling performance. This is further highlighted in our theoretical analysis as seen in Theorem 3.2. We also take a more nuanced look at the performance when the algorithm uses an upper confidence bound (with $C_1 = 0.25$) on the estimates and when it does not. We see the effects of not using any upper confidence bound (i.e. $C_1 = 0$). The left side tables in Tables 1,2 and 3 show the results when $C_1 = 0$ and the right side tables show the results when $C_1 = 0.25$. These show that not using UCB leads to high auto-labeling error (i.e. not meeting the guarantees) even when there is a sufficient amount of validation data. This can happen with high coverage as well—yielding a dataset with large errors. On the other hand using UCB, i.e. $C_1 = 0.25$, the algorithm can keep the auto-labeling error below the given threshold but suffers in coverage.

In the above Tables, we omitted AL, PL, and PL+SC for clarity. We provide full results with all baselines in Appendix D.3.

## 4.2 Role of Training Data Size

The labels queried for model training also play an important role in the process while incurring costs to obtain. The next experiment focuses on the impact of training data on auto-labeling.

| $N_q$ | Error (%) | | Coverage (%) | |
|---|---|---|---|---|
| | **TBAL** | **AL+SC** | **TBAL** | **AL+SC** |
| 200 | 1.67 $\pm$0.29 | 2.15 $\pm$0.45 | 73.30 $\pm$3.49 | 57.17 $\pm$11.09 |
| 400 | 1.63 $\pm$0.19 | 1.61 $\pm$0.29 | 72.59 $\pm$3.16 | 64.53 $\pm$16.61 |
| 600 | 1.67 $\pm$0.21 | 1.83 $\pm$0.30 | 71.38 $\pm$2.15 | 70.50 $\pm$5.68 |
| 800 | 1.67 $\pm$0.27 | 1.90 $\pm$0.31 | 69.10 $\pm$4.51 | 65.74 $\pm$10.14 |
| 1000 | 1.62 $\pm$0.22 | 1.97 $\pm$0.35 | 73.42 $\pm$2.84 | 68.05 $\pm$5.56 |

| $N_q$ | Error (%) | | Coverage (%) | |
|---|---|---|---|---|
| | **TBAL** | **AL+SC** | **TBAL** | **AL+SC** |
| 2000 | 9.22 $\pm$1.04 | 7.42 $\pm$0.71 | 17.51 $\pm$1.16 | 9.33 $\pm$0.66 |
| 4000 | 9.30 $\pm$0.38 | 6.97 $\pm$0.39 | 25.01 $\pm$1.20 | 14.25 $\pm$1.71 |
| 6000 | 9.12 $\pm$0.22 | 6.85 $\pm$0.26 | 28.06 $\pm$0.75 | 17.51 $\pm$0.36 |
| 8000 | 9.21 $\pm$0.14 | 7.38 $\pm$0.53 | 30.88 $\pm$0.64 | 21.18 $\pm$0.90 |
| 10000 | 8.95 $\pm$0.23 | 7.10 $\pm$0.26 | 32.31 $\pm$1.21 | 22.34 $\pm$0.61 |

Table 4: Results for varying $N_q$, the maximum number of samples algorithm can use for training. **Left: IMDB** with $N_v = 1000, C_1 = 0.25, \epsilon_a = 5\%$. **Right: Tiny-ImageNet** with $N_v = 10K, C_1 = 0.25, \epsilon_a = 10\%$. Mean and std. deviations are reported.

**Setup.** We limit the amount of training data the algorithm can use and record the resulting auto-labeling error and coverage. We ensure all algorithms have sufficiently large but equal amounts of validation data. We run on Unit-Ball, IMDB, Tiny-Imagenet, and CIFAR-10 datasets with the same values of $n_s$, $n_b$, and $C$ as in previous experiments.

**Results.** Figures 4(a), 4(b), and 2(b) indicate that TBAL and methods utilizing selective classification (AL+SC, PL+SC) maintain a high level of accuracy, even in scenarios where minimal training samples are used. This is expected as the threshold estimation method (when used with sufficient validation data) will find auto-labeling thresholds such that the auto-labeling error does not exceed $\epsilon_a$. The impact of training data size can be seen clearly in the coverage achieved by the algorithms. As expected, with fewer training samples the model has low accuracy leading to low coverage. However, as more samples are acquired, a more accurate model within the function class is learned, resulting in increased coverage. The Appendix D has additional discussion and results.

## 5 Related Work

We briefly review related work, deferring a more detailed discussion to the Appendix A.

There is a rich body of work on active learning (AL) [57, 12, 30, 28, 8] focused on learning the best model in a function class with less labeled data than passive learning. Various AL algorithms have been developed and analyzed, e.g., uncertainty sampling [62, 45], disagreement region based [9, 27], margin based [4] and abstention based methods that minimize the Chow's excess risk [72].

Selective classification (SC) equips a given classifier with the option to abstain from prediction in order to guarantee prediction quality. The foundations for SC are laid down in [17, 67, 18, 68] where results on the error rate in the prediction region and the coverage of the given classifier are provided. However, these works lack practical algorithms to find the prediction region. A recent work [26] gives a disagreement-based active learning strategy to learn a selective classifier.

A recent paper [49] studies a TBAL-like algorithm for auto-labeling. It focuses on the cost of training incurred when these systems use large-scale model classes for auto-labeling. It proposes an algorithm to predict the training set size that minimizes the overall cost and provides an empirical evaluation.

Weak supervision is another line of work aimed at auto-labeling that does not rely on obtaining human labels but instead uses potentially noisy but cheaply available sources to infer labels [51, 22]. In contrast, we are focused specifically on analyzing the performance of TBAL algorithms [58].

## 6 Conclusion and Future Work

In this work, we analyzed threshold-based auto-labeling systems and derived sample complexity bounds on the amount of human-labeled validation data required to guarantee the quality of machine-labeled data. Our study shows that these methods can accurately label a reasonable size of data using seemingly bad models when good confidence functions are available. Our analysis points to the hidden downside of these systems in terms of a large amount of validation data usage and calls for more sample-efficient methods including active testing.

## 7 Acknowledgments

This work was partly supported by funding from the American Family Data Science Institute. We thank Stephen Mussmann, Changho Shin, Albert Ge, Yi Chen, Kendall Park, and Nick Roberts for their valuable inputs. We thank the anonymous reviewers for their valuable comments and constructive feedback on our work.

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

# Supplementary Material

The supplementary material is organized as follows. We provide a detailed discussion on related works in Section A. We give details of the definitions and notation in Section B.1. The notations are also summarized in the Table 5 in section B. Then we give the proof of the main theorem (Theorem 3.2) followed by proofs of supporting lemmas in section C. We provide details of its instantiation for finite VC-dimension hypothesis classes and the homogeneous linear separators case in Section C.3. Then, we provide the technical details of the lower bound (Lemma 3.3). Then we provide details of additional experiments in Section D. In Section D.4 we provide insights into auto-labeling using PaCMAP [66] visualizations of auto-labeled regions in each round.

## A  Extended Related Work

There is a rich body of work on active learning on empirical and theoretical fronts [57, 12, 30, 28, 8, 54]. In active learning, the goal is to learn the best model in the given function class with fewer labeled data than in classical passive learning. To this end, various active learning algorithms have been developed and analyzed, e.g., uncertainty sampling [62, 45], disagreement region based [9, 27], margin based [3, 4], importance sampling based [5] and others [6]. Active learning has been shown to achieve exponentially smaller label complexity than passive learning in noiseless and low-noise settings [13, 3, 27, 28, 4, 11, 30, 6, 34, 32]. This suggests, in these settings auto-labeling using active learning followed by selective classification is expected to work well. However, in practice we do not have favorable noise conditions and the hypothesis class could be misspecified i.e. it may not contain the Bayes optimal classifier. In such cases, [31] proved lower bounds on the label complexity of active learning that are order wise same as passive learning. These findings have motivated more refined goals for active learning – abstain on hard to classify points and do well on the rest of the points. This idea is captured by the Chow's excess risk [7] and some of the recent works [60, 59, 47, 72] have proved exponential savings in label complexity for active learning when the goal is to minimize Chow's excess risk. The classifier learned by these methods is equipped with the abstain option and hence it can be readily applied for auto-labeling. However, the problem of misspecification of the hypothesis class still remains. Nevertheless, it would be interesting future work to explore the connections between auto-labeling and active learning with abstention. We also note that similar works on learning with abstention are done in the context of passive learning [10].

Another closely related line of work is the selective classification where the goal is to equip a given classifier with the option to abstain from the prediction in order to guarantee prediction quality. The foundations for selective classification are laid down in [17, 67, 18, 68] where they give results on the error rate in the prediction region and the coverage of a given classifier. However, they lack practical algorithms to find the prediction region. A recent work [26] proposes a new disagreement-based active learning strategy to learn a selective classifier.

Recent work studies a practical algorithm for threshold-based selective classification on deep neural networks [25]. The algorithm estimates the prediction threshold using training samples and they bound the error rate of the selective classifier using [23]. We note that their result is applicable to a specific setting of a given classifier. In contrast, in the TBAL algorithm analyzed in this paper, selective classification is done in each round and the classifiers are not given a priori but instead learned via ERM on training data which is adaptively sampled in each round.

Another related work [49] studies an algorithm similar to TBAL for auto-labeling. Their emphasis is on the cost of training incurred when these systems use large-scale model classes for auto-labeling. They propose an algorithm to predict the training set size that minimizes the overall cost and provides an empirical evaluation.

Well-calibrated uncertainty scores are essential to the success of threshold-based auto-labeling. However, in practice, such scores are often hard to get. Moreover, neural networks can produce overconfident ( unreliable) scores [29]. Fortunately, there are plenty of methods in the literature to deal with this problem [46, 69]. More recently, various approaches have been proposed for uncertainty calibration for neural networks [24, 42, 65, 35, 40, 56]. A detailed study of calibration methods and their impact on auto-labeling is beyond the scope of this work and left as future work.

There is another line of work emerging towards auto-labeling that does not rely on getting human labels but instead uses potentially noisy but cheaply available sources to infer labels [51, 52, 22]. The focus of this paper, however, is on analyzing the performance of TBAL algorithms [58, 2] that have emerged recently as auto-labeling solutions in systems.

# B  Definitions and Notation

## B.1  Basic Definitions

Let $\mathcal{X}$ be the instance space and $p(\mathbf{x})$ be a density function supported on $\mathcal{X}$. For any $\mathbf{x}_i \in \mathcal{X}$ let $y_i$ be its true label. Let $X = \{\mathbf{x}_1, \ldots, \mathbf{x}_N\}$ be a set of $N$ i.i.d samples drawn from $\mathcal{X}$. Let set $\mathcal{S} \subseteq \mathcal{X}$ denote a non-empty sub-region of $\mathcal{X}$ and $S \subseteq X \cap \mathcal{S}$ be a set of $n > 0$ i.i.d. samples.

**Definition B.1.** (Hypothesis Class with Abstain) We can think of the function $g$ along with set $T$ as inducing an extended hypothesis class $\mathcal{H}^{(T,g)}$. Let $\mathcal{H}^{T,g} = \mathcal{H} \times T$. For any function $(h,t) \in \mathcal{H}^{(T,g)}$ is defined as

$$(h,t)(\mathbf{x}) := \begin{cases} h(\mathbf{x}) & \text{if } g(h,\mathbf{x}) \geq t \\ \perp & \text{o.w.} \end{cases} \tag{3}$$

Here $(h,t)(\mathbf{x}) = \perp$ means the hypothesis $(h,t)$ abstains in classifying the point $\mathbf{x}$. Otherwise, it is equal to $h(\mathbf{x})$.

The subset $\mathcal{S}(h,t) \subseteq \mathcal{S}$ where $(h,t)$ does not abstain and its complement $\bar{\mathcal{S}}(h,t)$ where $(h,t)$ abstains, are defined as follows,

$$\mathcal{S}(h,t) := \{\mathbf{x} \in \mathcal{S} : (h,t)(\mathbf{x}) \neq \perp\}, \qquad \bar{\mathcal{S}}(h,t) := \{\mathbf{x} \in \mathcal{S} : (h,t)(\mathbf{x}) = \perp\}$$

**Probability Definitions.**   The probability $\mathbb{P}(\mathcal{S})$ of subset $\mathcal{S} \subseteq \mathcal{X}$ and the conditional probability of any subset $\mathcal{S}' \subseteq \mathcal{S}$ are given as follows,

$$\mathbb{P}(\mathcal{S}) := \mathbb{P}(\mathcal{S}|\mathcal{X}) := \int_{\mathbf{x} \in \mathcal{S}} p(\mathbf{x})d\mathbf{x}, \qquad \mathbb{P}(\mathcal{S}'|\mathcal{S}) := \frac{\mathbb{P}(\mathcal{S}'|\mathcal{X})}{\mathbb{P}(\mathcal{S}|\mathcal{X})}, \qquad \mathbb{P}(h,t|\mathcal{S}) := \mathbb{P}(\mathcal{S}(h,t)|\mathcal{S})$$

The empirical probabilities of $S$ and $S' \subseteq S$ are defined as follows,

$$\widehat{\mathbb{P}}(S) := \frac{|S|}{|X|}, \qquad \widehat{\mathbb{P}}(S'|S) := \frac{|S'|}{|S|}, \qquad \widehat{\mathbb{P}}(h,t|S) := \frac{|S(h,t)|}{|S|}$$

**Loss Functions.**   The loss functions are defined as follows,

$$\ell_{0-1}(h,\mathbf{x},y) := \mathbb{1}(h(\mathbf{x}) \neq y),$$
$$\ell_{\perp}(h,t,\mathbf{x}) := \mathbb{1}(g(h,\mathbf{x}) \geq t),$$
$$\ell_{0-1}^{\perp}(h,t,\mathbf{x},y) := \ell_{0-1}(h,\mathbf{x},y) \cdot \ell_{\perp}(h,t,\mathbf{x}).$$

**Error Definitions.**   Define the conditional error in set $\mathcal{S} \subseteq \mathcal{X}$ as follows,

$$\mathcal{E}(h,t|\mathcal{S}) := \mathbb{E}_{\mathbf{x}|\mathcal{S}}[\ell_{0-1}^{\perp}(h,t,\mathbf{x},y)] = \int_{\mathbf{x} \in \mathcal{S}} \ell_{0-1}^{\perp}(h,t,\mathbf{x},y) \cdot \frac{p(\mathbf{x})}{\mathbb{P}(\mathcal{S})}d\mathbf{x}$$

Then, the conditional error in set $\mathcal{S}(h,t)$ i.e. the subset of $\mathcal{S}$ on which $(h,t)$ does not abstain,

$$\mathcal{E}_a(h,t|\mathcal{S}) := \mathcal{E}(h,t|\mathcal{S}(h,t)) := \mathbb{E}_{\mathbf{x}|\mathcal{S}(h,t)}[\ell_{0-1}^{\perp}(h,t,\mathbf{x},y)] = \frac{\mathcal{E}(h,t|\mathcal{S})}{\mathbb{P}(h,t|\mathcal{S})}$$

Similarly, define their empirical counterparts as follows,

$$\widehat{\mathcal{E}}(h,t|S) := \frac{1}{|S|}\sum_{\mathbf{x}_i \in S} \ell_{0-1}^{\perp}(h,t,\mathbf{x}_i,y_i),$$

$$\widehat{\mathcal{E}}_a(h,t|S) := \widehat{\mathcal{E}}(h,t|S(h,t)) := \frac{1}{|S(h,t)|}\sum_{\mathbf{x}_i \in S(h,t)} \ell_{0-1}^{\perp}(h,t,\mathbf{x}_i,y_i),$$

Note that,

$$\sum_{\mathbf{x}_i \in S} \ell_{0-1}^{\perp}(h,t,\mathbf{x}_i,y_i) = \sum_{\mathbf{x}_i \in S(h,t)} \ell_{0-1}^{\perp}(h,t,\mathbf{x}_i,y_i) = \sum_{\mathbf{x}_i \in S(h,t)} \ell_{0-1}(h,\mathbf{x}_i,y_i)$$

| Symbol | Definition |
| --- | --- |
| $\mathcal{X}$ | feature space. |
| $\mathcal{Y}$ | label space. |
| $\mathcal{H}$ | hypothesis space. |
| $h$ | a hypothesis in $\mathcal{H}$. |
| $\mathbf{x}, y$ | $\mathbf{x}$ is an element in $\mathcal{X}$ and $y$ is its true label. |
| $\mathcal{S}, S$ | $\mathcal{S} \subseteq \mathcal{X}$ is a sub-region in $\mathcal{X}$, $S = \{\mathbf{x}_1, \ldots, \mathbf{x}_n\}$ i.i.d. samples in $\mathcal{S}$. |
| $X_{pool}$ | unlabeled pool of data points. |
| $X_v^{(i)}, n_v^{(i)}$ | set of validation points at the beginning of $i$th round and $n_v^{(i)} = |X_v^{(i)}|$. |
| $X_a^{(i)}, n_a^{(i)}$ | set of auto-labeled points in $i$th round and $n_a^{(i)} = |X_a^{(i)}|$. |
| $\hat{h}_i, \hat{t}_i$ | ERM solution and auto-labeling thresholds respectively in $i$th round. |
| $\mathcal{X}^{(i)}$ | unlabeled region left at the beginning of $i$th round. |
| $X^{(i)}$ | unlabeled pool left at the beginning of $i$th round. |
| $m_a^{(i)}$ | number of auto-labeling mistakes in $i$th round. |
| $k$ | number of rounds of the TBAL algorithm. |
| $X_{pool}(A_k)$ | set of all auto-labeled points till the end of round $k$. |
| $g$ | confidence function $g : \mathcal{H} \times \mathcal{X} \mapsto T$. Where $T \subseteq \mathbb{R}^+$, usually $T = [0, 1]$ |
| $\mathcal{H}^{T,g}$ | Cartesian product of $\mathcal{H}$ and $T$ the range of $g$. |
| $N_a^{(k)}$ | $\sum_{i=1}^{k} n_a^{(i)}$. |
| $\ell_{0-1}(h, \mathbf{x}, y)$ | $\mathbb{1}(h(\mathbf{x}) \neq y)$. |
| $\ell_{\perp}(h, t, \mathbf{x})$ | $\mathbb{1}(g(h, \mathbf{x}) \geq t)$. |
| $\ell_{0-1}^{\perp}(h, t, \mathbf{x}, y)$ | $\ell_{0-1}(h, \mathbf{x}, y) \cdot \ell_{\perp}((h, t), \mathbf{x})$. |
| $\mathfrak{R}_n(\mathcal{H}, \ell_{0-1})$ | $\mathbb{E}_{\sigma, S}\left[ \sup_{h \in \mathcal{H}} \frac{1}{n} \sum_{i=1}^{n} \sigma_i \ell_{0-1}(h, \mathbf{x}_i, y_i) \right]$. |
| $\mathfrak{R}_n(\mathcal{H}^{T,g}, \ell_{\perp})$ | $\mathbb{E}_{\sigma, S}\left[ \sup_{(h,t) \in \mathcal{H}^{T,g}} \frac{1}{n} \sum_{i=1}^{n} \sigma_i \ell_{\perp}(h, t, \mathbf{x}_i) \right]$. |
| $\mathfrak{R}_n(\mathcal{H}^{T,g}, \ell_{0-1}^{\perp})$ | $\mathbb{E}_{\sigma, S}\left[ \sup_{(h,t) \in \mathcal{H}^{T,g}} \frac{1}{n} \sum_{i=1}^{n} \sigma_i \ell_{0-1}^{\perp}(h, t, \mathbf{x}_i, y_i) \right]$. |
| $\mathfrak{R}_n(\mathcal{H}^{T,g})$ | $\mathfrak{R}_n(\mathcal{H}, \ell_{0-1}) + \mathfrak{R}_n(\mathcal{H}^{T,g}, \ell_{\perp})$. |
| $\mathcal{E}(h, t|\mathcal{S})$ | $\mathbb{E}_{\mathbf{x}|\mathcal{S}}[\ell_{0-1}^{\perp}(h, t, \mathbf{x}, y)]$. |
| $\widehat{\mathcal{E}}(h, t|S)$ | $\frac{1}{|S|} \sum_{i=1}^{|S|} \ell_{0-1}^{\perp}(h, t, \mathbf{x}_i, y_i)$. |
| $\mathbb{P}(h, t|\mathcal{S})$ | $\mathbb{E}_{\mathbf{x}|\mathcal{S}}[\ell_{\perp}(h, t, \mathbf{x})]$. |
| $\widehat{\mathbb{P}}(h, t|S)$ | $\frac{1}{|S|} \sum_{i=1}^{|S|} \ell_{\perp}(h, t, \mathbf{x}_i, y_i)$. |
| $\mathcal{E}_a(h, t|\mathcal{S})$ | $\mathcal{E}(h, t|\mathcal{S})/\mathbb{P}(h, t|\mathcal{S})$. |
| $\widehat{\mathcal{E}}_a(h, t|S)$ | $\widehat{\mathcal{E}}(h, t|S)/\widehat{\mathbb{P}}(h, t|S)$. |

Table 5: Glossary of variables and symbols used in this paper.

**Rademacher Complexity.** The Rademacher complexities for the function classes induced by the $\mathcal{H}, T, g$ and the loss functions are defined as follows,

$$\mathfrak{R}_n\big(\mathcal{H}, \ell_{0-1}\big) := \mathbb{E}_{\sigma, S}\Big[ \sup_{h \in \mathcal{H}} \frac{1}{n} \sum_{i=1}^{n} \sigma_i \ell_{0-1}(h, \mathbf{x}_i, y_i) \Big].$$

$$\mathfrak{R}_n\big(\mathcal{H}^{T,g}, \ell_{\perp}\big) := \mathbb{E}_{\sigma, S}\Big[ \sup_{(h,t) \in \mathcal{H}^{T,g}} \frac{1}{n} \sum_{i=1}^{n} \sigma_i \ell_{\perp}(h, t, \mathbf{x}_i) \Big].$$

$$\mathfrak{R}_n\big(\mathcal{H}^{T,g}, \ell_{0-1}^{\perp}\big) := \mathbb{E}_{\sigma, S}\Big[ \sup_{(h,t) \in \mathcal{H}^{T,g}} \frac{1}{n} \sum_{i=1}^{n} \sigma_i \ell_{0-1}^{\perp}(h, t, \mathbf{x}_i, y_i) \Big].$$

$$\mathfrak{R}_n\big(\mathcal{H}^{T,g}\big) := \mathfrak{R}_n\big(\mathcal{H}, \ell_{0-1}\big) + \mathfrak{R}_n\big(\mathcal{H}^{T,g}, \ell_{\perp}\big)$$

## B.2 Glossary

The notation is summarized in Table 5 below. More detailed notation is in section B.1.

## C Proofs

### C.1 Proofs for the General Setup

We begin by restating the theorem here and then give the proof.

**Theorem 3.2.** *(Overall Auto-Labeling Error and Coverage) Let $k$ denote the number of rounds of the TBAL Algorithm 1. Let $n_v^{(i)}, n_a^{(i)}$ denote the number of validation and auto-labeled points at epoch $i$ and $n^{(i)} = |X^{(i)}|$. Let $X_{pool}(A_k)$ be the set of auto-labeled points at the end of round $k$. $N_a^{(k)} = \sum_{i=1}^{k} n_a^{(i)}$ denote the total number of auto-labeled points. Then, for any $\delta \in (0, 1)$, with probability at least $1 - \delta$,*

$$
\widehat{\mathcal{E}}\big(X_{pool}(A_k)\big) \leq \sum_{i=1}^{k} \frac{n_a^{(i)}}{N_a^{(k)}} \left( \underbrace{\widehat{\mathcal{E}}_a\big(\hat{h}_i, \hat{t}_i | X_v^{(i)}\big)}_{(a)} + \underbrace{\frac{4}{p_0} \Big(\mathfrak{R}_{n_v^{(i)}}\big(\mathcal{H}^{T,g}\big) + \frac{2}{p_0}\sqrt{\frac{1}{n_v^{(i)}} \log(\frac{8k}{\delta})}\Big)}_{(b)} \right)
$$

$$
+ \frac{4}{p_0}\underbrace{\left( \sum_{i=1}^{k} \frac{n_a^{(i)}}{N_a^{(k)}} \mathfrak{R}_{n_a^{(i)}}\big(\mathcal{H}^{T,g}\big) + \sqrt{\frac{k}{N_a^{(k)}} \log(\frac{8k}{\delta})} \right)}_{(c)}, \quad and
$$

$$
\widehat{\mathcal{P}}(X_{pool}(A_k)) \geq \sum_{i=1}^{k} \mathbb{P}\big(\mathcal{X}^{(i)}(\hat{h}_i, \hat{t}_i)\big) - 2\mathfrak{R}_{n^{(i)}}\big(\mathcal{H}^{T,g}\big) - \sqrt{\frac{2k^2}{N} \log\left(\frac{8k}{\delta}\right)}.
$$

*Proof.* Recall the definition of auto-labeling error,

$$
\widehat{\mathcal{E}}\big(X_{pool}(A_k)\big) = \sum_{i=1}^{k} \frac{m_a^{(i)}}{N_a^{(k)}}, \qquad m_a^{(i)} = n_a^{(i)} \cdot \widehat{\mathcal{E}}_a\big(\hat{h}_i, \hat{t}_i | X^{(i)}\big).
$$

Here, $m_a^{(i)}$ is the number of auto-labeling mistakes made by the Algorithm in the $i$th round and $\widehat{\mathcal{E}}_a\big(\hat{h}_i, \hat{t}_i | X^{(i)}\big)$ is the auto-labeling error in that round. Note that we cannot observe these quantities since the true labels for the auto-labeled points are not available. To estimate the auto-labeling error of each round we make use of validation data. Using the validation data we first get an upper bound on the true error rate of the auto-labeling region i.e. $\mathcal{E}_a\big(\hat{h}_i, \hat{t}_i | \mathcal{X}^{(i)}\big)$ in terms of the auto-labeling error on the validation data $\widehat{\mathcal{E}}_a\big(\hat{h}_i, \hat{t}_i | X_v^{(i)}\big)$ and then get an upper bound on empirical auto-labeling error rate $\widehat{\mathcal{E}}_a\big(\hat{h}_i, \hat{t}_i | X^{(i)}\big)$ using the true error rate of the auto-labeling region.

We get these bounds by application of Lemma C.1 with $\delta_3 = \delta/4k$ for each round and then apply union bound over all $k$ epochs. Note that we have to apply the lemma twice, first to get the concentration bound w.r.t the validation data and second to get the concentration w.r.t to the auto-labeled points.

$$
\mathcal{E}_a\big(\hat{h}_i, \hat{t}_i | \mathcal{X}^{(i)}\big) \leq \widehat{\mathcal{E}}_a\big(\hat{h}_i, \hat{t}_i | X_v^{(i)}\big) + \frac{4}{p_0}\mathfrak{R}_{n_v^{(i)}}\big(\mathcal{H}^{T,g}\big) + \frac{2}{p_0}\sqrt{\frac{1}{n_v^{(i)}} \log\left(\frac{8k}{\delta}\right)}.
$$

$$
\widehat{\mathcal{E}}_a\big(\hat{h}_i, \hat{t}_i | X^{(i)}\big) \leq \mathcal{E}_a\big(\hat{h}_i, \hat{t}_i | \mathcal{X}^{(i)}\big) + \frac{4}{p_0}\mathfrak{R}_{n_a^{(i)}}\big(\mathcal{H}^{T,g}\big) + \frac{2}{p_0}\sqrt{\frac{1}{n_a^{(i)}} \log\left(\frac{8k}{\delta}\right)}.
$$

Substituting $\mathcal{E}_a\big(\hat{h}_i, \hat{t}_i | \mathcal{X}^{(i)}\big)$ by its upper confidence bound on the validation data.

$$
\widehat{\mathcal{E}}_a\big(\hat{h}_i, \hat{t}_i | X^{(i)}\big) \leq \widehat{\mathcal{E}}_a\big(\hat{h}_i, \hat{t}_i | X_v^{(i)}\big) + \frac{2}{p_0}\mathfrak{R}_{n_v^{(i)}}\big(\mathcal{H}^{T,g}\big) + \frac{2}{p_0}\mathfrak{R}_{n_a^{(i)}}\big(\mathcal{H}^{T,g}\big)
$$

$$
+ \frac{2}{p_0}\sqrt{\frac{1}{n_v^{(i)}} \log\left(\frac{8k}{\delta}\right)} + \frac{2}{p_0}\sqrt{\frac{1}{n_a^{(i)}} \log\left(\frac{8k}{\delta}\right)}.
$$

Having an upper bound on the empirical auto-labeling error for $i^{th}$ round gives us an upper bound on the number of auto-labeling mistakes $m_a^{(i)}$ made in that round. It allows us to upper bound the total auto-labeling mistakes in all $k$ rounds and thus the overall auto-labeling error as detailed below,

$$\widehat{\mathcal{E}}\big(X_{pool}(A_k)\big) = \sum_{i=1}^{k} \frac{m_a^{(i)}}{N_a^{(k)}}, \qquad m_a^{(i)} = n_a^{(i)} \cdot \widehat{\mathcal{E}}_a\big(\hat{h}_i, \hat{t}_i | X^{(i)}\big).$$

Since we have an upper bound on the empirical auto-labeling error in each round, we have an upper bound for each $m_a^{(i)}$, which are used as follows to get the bound on the auto-labeling error,

$$
\begin{aligned}
\widehat{\mathcal{E}}(X_{pool}(A_k)) &= \sum_{i=1}^{k} \frac{m_a^{(i)}}{N_a^{(k)}} \\
&= \sum_{i=1}^{k} \frac{n_a^{(i)}}{N_a^{(k)}} \cdot \frac{m_a^{(i)}}{n_a^{(i)}} \\
&= \sum_{i=1}^{k} \frac{n_a^{(i)}}{N_a^{(k)}} \cdot \widehat{\mathcal{E}}_a\big(\hat{h}_i, \hat{t}_i | X^{(i)}\big) \\
&\leq \sum_{i=1}^{k} \frac{n_a^{(i)}}{N_a^{(k)}} \cdot \left( \mathcal{E}_a\big(\hat{h}_i, \hat{t}_i | \mathcal{X}^{(i)}\big) + \frac{4}{p_0} \mathfrak{R}_{n_a^{(i)}}\big(\mathcal{H}^{T,g}\big) + \frac{2}{p_0}\sqrt{\frac{1}{n_a^{(i)}}\log\left(\frac{8k}{\delta}\right)} \right) \\
&\leq \sum_{i=1}^{k} \frac{n_a^{(i)}}{N_a^{(k)}} \cdot \left( \widehat{\mathcal{E}}_a\big(\hat{h}_i, \hat{t}_i | X_v^{(i)}\big) + \frac{4}{p_0} \mathfrak{R}_{n_v^{(i)}}\big(\mathcal{H}^{T,g}\big) + \frac{2}{p_0}\sqrt{\frac{1}{n_v^{(i)}}\log\left(\frac{8k}{\delta}\right)} \right. \\
&\qquad\qquad \left. + \frac{4}{p_0} \mathfrak{R}_{n_a^{(i)}}\big(\mathcal{H}^{T,g}\big) + \frac{2}{p_0}\sqrt{\frac{1}{n_a^{(i)}}\log\left(\frac{8k}{\delta}\right)} \right) \\
&\leq \sum_{i=1}^{k} \frac{n_a^{(i)}}{N_a^{(k)}} \cdot \left( \widehat{\mathcal{E}}_a\big(\hat{h}_i, \hat{t}_i | X_v^{(i)}\big) + \frac{4}{p_0} \mathfrak{R}_{n_v^{(i)}}\big(\mathcal{H}^{T,g}\big) + \frac{4}{p_0} \mathfrak{R}_{n_a^{(i)}}\big(\mathcal{H}^{T,g}\big) + \right. \\
&\qquad\qquad \left. \frac{2}{p_0}\sqrt{\frac{1}{n_v^{(i)}}\log\left(\frac{8k}{\delta}\right)} \right) + \sum_{i=1}^{k} \frac{n_a^{(i)}}{N_a^{(k)}} \cdot \left( \frac{2}{p_0}\sqrt{\frac{1}{n_a^{(i)}}\log\left(\frac{8k}{\delta}\right)} \right)
\end{aligned}
$$

The last term is simplified as follows,

$$
\begin{aligned}
\sum_{i=1}^{k} \frac{n_a^{(i)}}{N_a^{(k)}} \cdot \left( \frac{2}{p_0}\sqrt{\frac{1}{n_a^{(i)}}\log\left(\frac{8k}{\delta}\right)} \right) &= \frac{2}{p_0} \cdot \sum_{i=1}^{k} \frac{n_a^{(i)}}{N_a^{(k)}} \sqrt{\frac{1}{n_a^{(i)}}\log\left(\frac{8k}{\delta}\right)} \\
&= \frac{2}{p_0} \cdot \frac{1}{N_a^{(k)}} \sum_{i=1}^{k} \sqrt{n_a^{(i)} \log\left(\frac{8k}{\delta}\right)} \\
&= \frac{2}{p_0} \cdot \sqrt{\log\left(\frac{8k}{\delta}\right)} \cdot \sum_{i=1}^{k} \frac{\sqrt{n_a^{(i)}}}{N_a^{(k)}} \\
&\leq \frac{2}{p_0} \cdot \sqrt{\log\left(\frac{8k}{\delta}\right)} \cdot \sqrt{\frac{k}{N_a^{(k)}}} \\
&= \frac{2}{p_0} \cdot \sqrt{\frac{k}{N_a^{(k)}} \log\left(\frac{8k}{\delta}\right)}
\end{aligned}
$$

The last inequality follows from the application of the inequality $||\mathbf{u}||_1 \leq \sqrt{k}||\mathbf{u}||_2$ for any vector $\mathbf{u} \in \mathbb{R}^k$. Here we let $\mathbf{u} = [\sqrt{n_a^{(1)}}, \ldots, \sqrt{n_a^{(k)}}]$, and since $\forall i \ \sqrt{n_a^{(i)}} > 0$ so, $\sum_{i=1}^{k} \sqrt{n_a}^{(i)} = ||\mathbf{u}||_1$

and $N_a^{(k)} = ||\mathbf{u}||_2^2$.

$$\frac{\sum_{i=1}^{k} \sqrt{n_a^{(i)}}}{N_a^{(k)}} = \frac{||\mathbf{u}||_1}{||\mathbf{u}||_2^2} \leq \frac{\sqrt{k}||\mathbf{u}||_2}{||\mathbf{u}||_2^2} = \frac{\sqrt{k}}{||\mathbf{u}||_2} = \sqrt{\frac{k}{N_a^{(k)}}}$$

To get the bound on coverage we follow the same steps except that we can use all the unlabeled pool of size $n^{(i)}$ to estimate the coverage in each round which gives us the bound in terms of $n^{(i)}$ and $N$ as follows,

$$\begin{aligned}
\widehat{\mathcal{P}}(X_{pool}(A_k)) &= \frac{1}{N} \sum_{i=1}^{k} n_a^{(i)} \\
&= \frac{1}{N} \sum_{i=1}^{k} n^{(i)} \cdot \frac{n_a^{(i)}}{n^{(i)}} \\
&= \frac{1}{N} \sum_{i=1}^{k} n^{(i)} \cdot \widehat{\mathbb{P}}(X_a^{(i)}|X^{(i)}) \\
&= \frac{1}{N} \sum_{i=1}^{k} n^{(i)} \cdot \widehat{\mathbb{P}}(\hat{h}_i, \hat{t}_i|X^{(i)}) \\
&\geq \sum_{i=1}^{k} \frac{n^{(i)}}{N} \left( \mathbb{P}(\hat{h}_i, \hat{t}_i|\mathcal{X}^{(i)}) - 2\Re_{n^{(i)}}(\mathcal{H}^{T,g}) - \sqrt{\frac{1}{n^{(i)}} \log\left(\frac{8k}{\delta}\right)} \right) \\
&\geq \sum_{i=1}^{k} \frac{n^{(i)}}{N} \left( \mathbb{P}(\hat{h}_i, \hat{t}_i|\mathcal{X}^{(i)}) - 2\Re_{n^{(i)}}(\mathcal{H}^{T,g}) \right) - \sqrt{\frac{k}{N} \log\left(\frac{8k}{\delta}\right)}
\end{aligned}$$

We bound the first term as follows,

$$\begin{aligned}
\sum_{i=1}^{k} \frac{n^{(i)}}{N} \mathbb{P}(\hat{h}_i, \hat{t}_i|\mathcal{X}^{(i)}) &= \sum_{i=1}^{k} \frac{n^{(i)}}{N} \cdot \frac{\mathbb{P}(\mathcal{X}^{(i)}(\hat{h}_i, \hat{t}_i))}{\mathbb{P}(\mathcal{X}^{(i)})} \\
&\geq \sum_{i=1}^{k} \left( \mathbb{P}(\mathcal{X}^{(i)}) - \sqrt{\frac{1}{N} \log\left(\frac{8k}{\delta}\right)} \right) \cdot \left( \frac{\mathbb{P}(\mathcal{X}(\hat{h}_i, \hat{t}_i))}{\mathbb{P}(\mathcal{X}^{(i)})} \right) \\
&\geq \sum_{i=1}^{k} \mathbb{P}(\mathcal{X}^{(i)}(\hat{h}_i, \hat{t}_i)) - \sqrt{\frac{1}{N} \log\left(\frac{8k}{\delta}\right)}
\end{aligned}$$

Substituting it back we get,

$$\begin{aligned}
\widehat{\mathcal{P}}(X_{pool}(A_k)) &\geq \sum_{i=1}^{k} \mathbb{P}(\mathcal{X}^{(i)}(\hat{h}_i, \hat{t}_i)) - 2\Re_{n^{(i)}}(\mathcal{H}^{T,g}) - k\sqrt{\frac{1}{N} \log\left(\frac{8k}{\delta}\right)} - \sqrt{\frac{k}{N} \log\left(\frac{8k}{\delta}\right)} \\
&\geq \sum_{i=1}^{k} \mathbb{P}(\mathcal{X}^{(i)}(\hat{h}_i, \hat{t}_i)) - 2\Re_{n^{(i)}}(\mathcal{H}^{T,g}) - \sqrt{\frac{4k^2}{N} \log\left(\frac{8k}{\delta}\right)}
\end{aligned}$$

For the last step we use the inequality $\sqrt{a} + \sqrt{b} \leq \sqrt{2(a+b)}$ for any $a, b \in \mathbb{R}^+$. $\qquad\square$

**Rademacher complexity and validation error trade-off.** The bound contains validation errors at different thresholds. We revisit the definition of validation error appearing in the bound. Let $X_v = \{x_1, \ldots x_{n_v}\}$ be the validation samples and $y_i$ be the label corresponding to $x_i$. Given any $h \in \mathcal{H}$ and the confidence function $g$, we have different subsets $X_v(h, t)$ of the validation points for

which the model's confidence is higher than $t$. More precisely, $X_v(h, t) = \{x_i \in X_v : g(h, x_i) \geq t\}$ and the validation error $\hat{\mathcal{E}}_a(h, t|X_v)$ is computed on each of these subsets as follows, $\hat{\mathcal{E}}_a(h, t|X_v) = \frac{1}{|X_v(h,t)|} \sum_{x_i \in X_v(h,t)} \mathbf{1}(h(x_i) \neq y_i)$ this error is different from the overall validation error which is computed over the entire set of validation points $X_v$:

$$\hat{\mathcal{E}}(h|X_v) = \frac{1}{|X_v|} \sum_{x_i \in X_v} \mathbf{1}(h(x_i) \neq y_i).$$

The TBAL method computes $\hat{\mathcal{E}}_a(h, t|X_v)$ at different thresholds ($t$) and selects the threshold at which it is at most $\epsilon$. Thus even if the overall validation error $\hat{\mathcal{E}}(h|X_v)$ is bad, there could still be regions in the space where the conditional validation error $\hat{\mathcal{E}}_a(h, t|X_v)$ is small. This can be easily seen in Figure 2 in the paper. Here we are doing auto-labeling using a linear function class that has low Rademacher complexity. All the models in this class have high overall validation error $\hat{\mathcal{E}}(h|X_v)$ but there are subsets where the conditional validation error $\hat{\mathcal{E}}_a(h, t|X_v)$ is small and TBAL is able to find those subsets. Thus we see that working with low Rademacher complexity classes might lead to high overall validation error. However, it does not affect TBAL as long as there are regions of low conditional validation error $\hat{\mathcal{E}}_a(h, t|X_v)$. Furthermore, our upper bound on excess auto-labeling error depends on $\hat{\mathcal{E}}_a(h, t|X_v)$ which due to the TBAL procedure is at most $\epsilon$ and hence it is not in conflict with the Rademacher complexity term.

Next, we state the result for uniform convergence between $\mathcal{E}_a(h, t|\mathcal{S}), \hat{\mathcal{E}}_a(h, t|S)$ and give its proof.

**Lemma C.1.** *For any $\delta_3, p_0 \in (0, 1)$, let $\mathcal{S}$ and $S$ be defined as above. Let $\mathbb{P}(h, t|\mathcal{S}) \geq p_0$ and $\hat{\mathbb{P}}(h, t|S) \geq p_0 \, \forall (h, t) \in \mathcal{H}^{T,g}$, the following holds w.p. at least $1 - \delta_3/2$*

$$\left| \mathcal{E}_a(h, t|\mathcal{S}) - \hat{\mathcal{E}}_a(h, t|S) \right| \leq \frac{4}{p_0} \mathfrak{R}_n(\mathcal{H}^{T,g}) + \frac{2}{p_0} \sqrt{\frac{1}{n} \log(\frac{2}{\delta_3})} \qquad \forall (h, t) \in \mathcal{H}^{T,g}. \qquad (4)$$

*Proof.* We begin with proving one side of the inequality and the other side is shown by following the same steps. The proof is based on applying the uniform convergence results for $\hat{\mathcal{E}}(h, t|S)$ and $\hat{\mathbb{P}}(h, t|S)$ from Lemma C.2. The main difficulty here is that $\mathbb{E}_S[\hat{\mathcal{E}}_a(h, t|S)] \neq \mathcal{E}_a(h, t|\mathcal{S})$, so we cannot directly get the above result from standard uniform convergence bounds.

We prove it, by using the results from the Lemma C.2 and restricting the region $\mathcal{S}$ such that it has probability mass at least $p_0$.

By definitions of $\mathcal{E}_a(h, t|\mathcal{S})$ and $\hat{\mathcal{E}}_a(h, t|S)$ we have,

$$\mathcal{E}(h, t|\mathcal{S}) = \mathbb{P}(h, t|\mathcal{S}) \cdot \mathcal{E}_a(h, t|\mathcal{S}) \qquad \text{and} \qquad \hat{\mathcal{E}}(h, t|S) = \hat{\mathbb{P}}(h, t|S) \cdot \hat{\mathcal{E}}_a(h, t|S).$$

Let $\xi_1 = \sqrt{(1/n) \log(2/\delta_1)}, \xi_2 = \sqrt{(1/n) \log(2/\delta_2)}$. From lemma C.2 we have,

$$\mathcal{E}(h, t|\mathcal{S}) \leq \hat{\mathcal{E}}(h, t|S) + 2\mathfrak{R}_n(\mathcal{H}^{T,g}) + \xi_1 \quad \forall (h, t) \in \mathcal{H}^{T,g} \quad \text{w.p. } 1 - \delta_1/2. \qquad (5)$$

$$\hat{\mathbb{P}}(h, t|S) \leq \mathbb{P}(h, t|\mathcal{S}) + 2\mathfrak{R}_n(\mathcal{H}^{T,g}) + \xi_2 \quad \forall (h, t) \in \mathcal{H}^{T,g} \quad \text{w.p. } 1 - \delta_2/2. \qquad (6)$$

Plugging in the above definitions of errors in equation (5) we get,

$$\mathbb{P}(h, t|\mathcal{S}) \cdot \mathcal{E}_a(h, t|\mathcal{S}) \leq \hat{\mathbb{P}}(h, t|S) \cdot \hat{\mathcal{E}}_a(h, t|S) + 2\mathfrak{R}_n(\mathcal{H}^{T,g}) + \xi_1. \qquad (7)$$

$$\mathcal{E}_a(h, t|\mathcal{S}) \leq \frac{\hat{\mathbb{P}}(h, t|S)}{\mathbb{P}(h, t|\mathcal{S})} \hat{\mathcal{E}}_a(h, t|S) + 2\frac{\mathfrak{R}_n(\mathcal{H}^{T,g})}{\mathbb{P}(h, t|\mathcal{S})} + \frac{\xi_1}{\mathbb{P}(h, t|\mathcal{S})}. \qquad (8)$$

Substituting $\hat{\mathbb{P}}(h, t|S)$ from equation 6 in the above equation, we get the following w.p. $(1 - \delta_1/2)(1 - \delta_2/2), \forall (h, t) \in \mathcal{H}^{T,g}$ ,

$$\mathcal{E}_a(h, t|\mathcal{S}) \leq \left( \frac{\mathbb{P}(h, t|\mathcal{S}) + 2\mathfrak{R}_n(\mathcal{H}^{T,g}) + \xi_2}{\mathbb{P}(h, t|\mathcal{S})} \right) \hat{\mathcal{E}}_a(h, t|S) + \frac{2\mathfrak{R}_n(\mathcal{H}^{T,g})}{\mathbb{P}(h, t|\mathcal{S})} + \frac{\xi_1}{\mathbb{P}(h, t|\mathcal{S})} \qquad .$$

$$= \left(1 + \frac{2\Re_n(\mathcal{H}^{T,g})}{\mathbb{P}(h,t|\mathcal{S})} + \frac{\xi_2}{\mathbb{P}(h,t|\mathcal{S})}\right)\widehat{\mathcal{E}}_a(h,t|S) + \frac{2\Re_n(\mathcal{H}^{T,g})}{\mathbb{P}(h,t|\mathcal{S})} + \frac{\xi_1}{\mathbb{P}(h,t|\mathcal{S})}.$$

$$= \widehat{\mathcal{E}}_a(h,t|S) + \frac{2\Re_n(\mathcal{H}^{T,g})}{\mathbb{P}(h,t|\mathcal{S})} \cdot \widehat{\mathcal{E}}_a(h,t|S) + \frac{\xi_2}{\mathbb{P}(h,t|\mathcal{S})}\widehat{\mathcal{E}}_a(h,t|S) + \frac{2\Re_n(\mathcal{H}^{T,g})}{\mathbb{P}(h,t|\mathcal{S})}$$
$$+ \frac{\xi_1}{\mathbb{P}(h,t|\mathcal{S})}.$$

Using upper bound $\widehat{\mathcal{E}}_a(h,t|S) \leq 1$ in the second and third terms,

$$\mathcal{E}_a(h,t|\mathcal{S}) \leq \widehat{\mathcal{E}}_a(h,t|S) + 4\frac{\Re_n(\mathcal{H}^{T,g})}{\mathbb{P}(h,t|\mathcal{S})} + \frac{\xi_1 + \xi_2}{\mathbb{P}(h,t|\mathcal{S})} \qquad \forall(h,t) \in \mathcal{H}^{T,g} \; w.p. \geq 1 - (\delta_1 + \delta_2)/2.$$

Using $\mathbb{P}(h,t|\mathcal{S}) \geq p_0$

$$\mathcal{E}_a(h,t|\mathcal{S}) \leq \widehat{\mathcal{E}}_a(h,t|S) + \frac{4}{p_0}\Re_n(\mathcal{H}^{T,g}) + \frac{\xi_1 + \xi_2}{p_0} \qquad \forall(h,t) \in \mathcal{H}^{T,g} \; w.p. \geq 1 - (\delta_1 + \delta_2)/2.$$

Letting $\delta_1 = \delta_2 = \delta_3$ and $\xi_1 = \xi_2 = \frac{\xi \cdot p_0}{2}$ gives $\xi = \sqrt{\frac{4}{p_0^2 n} \log\left(\frac{2}{\delta_3}\right)}$ and

$$\mathcal{E}_a(h,t|\mathcal{S}) \leq \widehat{\mathcal{E}}_a(h,t|S) + \frac{4}{p_0}\Re_n(\mathcal{H}^{T,g}) + \xi \qquad \forall(h,t) \in \mathcal{H}^{T,g} \; w.p. \geq 1 - \delta_3.$$

This proves one side of the result, and the other side of the result follows similarly. $\qquad\square$

**Lemma C.2.** *Let $\mathcal{S} \subseteq \mathcal{X}$ be a sub-region of $\mathcal{X}$ and $S = \{\mathbf{x}_1, \ldots, \mathbf{x}_n\}$ be a set of $n$ i.i.d samples in $\mathcal{S}$ drawn from distribution $P_{\mathbf{x}}$. Let $\{y_1, \ldots y_n\}$ be the corresponding true labels, let $\Re_n(\mathcal{H}^{T,g})$ be the rademacher complexity of class $\mathcal{H}^{T,g}$ then for any $\delta_1, \delta_2 \in (0,1)$ we have,*

$$|\mathcal{E}(h,t|\mathcal{S}) - \widehat{\mathcal{E}}(h,t|S)| \leq 2\Re_n(\mathcal{H}^{T,g}) + \sqrt{\frac{1}{n}\log(\frac{2}{\delta_1})} \quad \forall(h,t) \in \mathcal{H}^{T,g} \quad w.p. \; 1 - \delta_1/2. \qquad (9)$$

$$|\mathbb{P}(h,t|\mathcal{S}) - \hat{\mathbb{P}}(h,t|S)| \leq 2\Re_n(\mathcal{H}^{T,g}) + \sqrt{\frac{1}{n}\log(\frac{2}{\delta_2})} \quad \forall(h,t) \in \mathcal{H}^{T,g} \quad w.p. \; 1 - \delta_2/2. \qquad (10)$$

*Proof.* The proof is similar to the standard proofs for Rademacher complexity based generalization error bound. Since we work with the modified loss function and hypothesis class to include the abstain option, for completeness we give the proof here. The proofs for error and probability bounds are very much the same except for the change in the loss function. We give the proof for the error bound here.

The result follows by applying McDiarmid's inequality on the function $\phi(S)$ defined as below,

$$\phi(S) := \sup_{(h,t) \in \mathcal{H}^{T,g}} \mathcal{E}(h,t|\mathcal{S}) - \widehat{\mathcal{E}}_S(h,t|S).$$

To apply McDiarmid's inequality we first show that $\phi(S)$ satisfies the bounded difference property (Lemma C.4). This gives us,

$$\mathcal{E}(h,t|\mathcal{S}) - \widehat{\mathcal{E}}_S(h,t|S) \leq \phi(S) \leq \mathbb{E}_S[\phi(S)] + \sqrt{\frac{1}{n}\log(\frac{2}{\delta_1})} \quad \forall(h,t) \in \mathcal{H}^{T,g} \quad w.p. \; 1 - \frac{\delta_1}{2}.$$

Using the bound on $\mathbb{E}_S[\phi(S)]$ from Lemma C.3 we get,

$$\mathcal{E}(h,t|\mathcal{S}) \leq \widehat{\mathcal{E}}(h,t|S) + 2\Re_n(\mathcal{H}^{T,g}) + \sqrt{\frac{1}{n}\log(\frac{2}{\delta_1})} \quad \forall(h,t) \in \mathcal{H}^{T,g} \quad w.p. \; 1 - \frac{\delta_1}{2}.$$

Similarly, the bound for the other side is obtained which holds w.p. $1 - \delta_1/2$, and combining both we get eq. (9).

The bound of probabilities is obtained by following the same steps as above but with a different loss function, $\ell_\perp$, since $\mathbb{P}(h,t|\mathcal{S})$ is the probability mass of the region where $(h,t)$ does not abstain.

$\qquad\square$

**Lemma C.3.** *Let $\mathcal{S} \subseteq \mathcal{X}$ be a sub-region of $\mathcal{X}$ and $S = \{\mathbf{x}_1, \ldots, \mathbf{x}_n\}$ be a set of $n$ i.i.d samples in $\mathcal{S}$ drawn from distribution $P_{\mathbf{x}}$. Let $\{y_1, \ldots y_n\}$ be the corresponding true labels and let $\mathfrak{R}_n(\mathcal{H}^{T,g})$ be the Rademacher complexity of the function class $\mathcal{H}^{T,g}$ defined over $n$ i.i.d. samples. Then we have,*

$$\mathbb{E}_S\Big[\sup_{(h,t)\in\mathcal{H}^{T,g}} \mathcal{E}(h,t|\mathcal{S}) - \widehat{\mathcal{E}}(h,t|S)\Big] \leq 2\mathfrak{R}_n(\mathcal{H}^{T,g}). \tag{11}$$

*Proof.* Let $\tilde{S} = \{\tilde{\mathbf{x}}_1, \tilde{\mathbf{x}}_2, \ldots \tilde{\mathbf{x}}_n\}$ be another set of independent draws from the same distribution as of $S$ and let the corresponding labels be $\{\tilde{y}_1, \ldots \tilde{y}_n\}$. These samples are usually termed as *ghost samples* and do not need to be counted in the sample complexity.

$$\mathbb{E}_S\Big[\sup_{(h,t)\in\mathcal{H}^{T,g}} \mathcal{E}(h,t|\mathcal{S}) - \widehat{\mathcal{E}}(h,t|S)\Big] = \mathbb{E}_S\Big[\sup_{(h,t)\in\mathcal{H}^{T,g}} \mathbb{E}_{\tilde{S}}\big[\widehat{\mathcal{E}}(h,t|\tilde{S})\big] - \widehat{\mathcal{E}}(h,t|S)\Big].$$

$$= \mathbb{E}_S\Big[\sup_{(h,t)\in\mathcal{H}^{T,g}} \mathbb{E}_{\tilde{S}}\big[\widehat{\mathcal{E}}(h,t|\tilde{S}) - \widehat{\mathcal{E}}(h,t|S)\big]\Big].$$

$$\leq \mathbb{E}_S\Big[\mathbb{E}_{\tilde{S}}\Big[\sup_{(h,t)\in\mathcal{H}^{T,g}} \big[\widehat{\mathcal{E}}(h,t|\tilde{S}) - \widehat{\mathcal{E}}(h,t|S)\big]\Big]\Big].$$

$$= \mathbb{E}_{S,\tilde{S}}\Big[\sup_{h\in\mathcal{H}^{T,g}} \big[\widehat{\mathcal{E}}(h,t|\tilde{S}) - \widehat{\mathcal{E}}(h,t|S)\big]\Big].$$

$$= \mathbb{E}_{S,\tilde{S}}\Big[\sup_{(h,t)\in\mathcal{H}^{T,g}} \Big[\frac{1}{n}\sum_{i=1}^{n}\ell_{0-1}^{\perp}(h,t,\tilde{\mathbf{x}}_i,\tilde{y}_i) - \frac{1}{n}\sum_{i=1}^{n}\ell_{0-1}^{\perp}(h,t,\mathbf{x}_i,y_i)\Big]\Big].$$

$$= \mathbb{E}_{\sigma,S,\tilde{S}}\Big[\sup_{h\in\mathcal{H}^{T,g}} \Big[\frac{1}{n}\sum_{i=1}^{n}\sigma_i\ell_{0-1}^{\perp}(h,t,\tilde{\mathbf{x}}_i,\tilde{y}_i) - \frac{1}{n}\sum_{i=1}^{n}\sigma_i\ell_{0-1}^{\perp}(h,t,\mathbf{x}_i,y_i)\Big]\Big].$$

$$\leq \mathbb{E}_{\sigma,\tilde{S}}\Big[\sup_{(h,t)\in\mathcal{H}^{T,g}} \frac{1}{n}\sum_{i=1}^{n}\sigma_i\ell_{0-1}^{\perp}(h,t,\tilde{\mathbf{x}}_i,\tilde{y}_i)\Big] +$$

$$\mathbb{E}_{\sigma,S}\Big[\sup_{(h,t)\in\mathcal{H}^{T,g}} \frac{1}{n}\sum_{i=1}^{n}\sigma_i\ell_{0-1}^{\perp}(h,t,\mathbf{x}_i,y_i)\Big].$$

$$= 2\mathfrak{R}_n(\mathcal{H}^{T,g}, \ell_{0-1}^{\perp}).$$

$$\leq 2\mathfrak{R}_n(\mathcal{H}^{T,g}).$$

In the last step, we used the upper bound on the Rademacher complexity from Lemma C.5. $\qquad\square$

**Lemma C.4.** *(Bounded Difference) Let $S$ be a set of i.i.d samples from $P_{\mathbf{x}}$ then for $\phi(S) := \sup_{(h,t)\in\mathcal{H}^{T,g}} \mathcal{E}(h,t|\mathcal{S}) - \widehat{\mathcal{E}}(h,t|S)$, with probability at least $1 - \delta$,*

$$\phi(S) \leq \mathbb{E}_S[\phi(S)] + \sqrt{\frac{1}{|S|}\log(\frac{1}{\delta})} \tag{12}$$

*Proof.* It is proved by showing that $\phi(S)$ satisfies the conditions (in particular the bounded difference assumption) needed for the application of McDiarmid Inequality. To see this, Let $S = \{\mathbf{x}_1, \mathbf{x}_2, \ldots \mathbf{x}_i, \ldots, \mathbf{x}_n\}$ and let $S' = \{\mathbf{x}_1, \mathbf{x}_2, \ldots \mathbf{x}_i', \ldots, \mathbf{x}_n\}$, i.e. $S$ and $S'$ may differ only on the $i^{th}$ sample.

$$|\phi(S) - \phi(S')| = \Big|\sup_{(h,t)\in\mathcal{H}^{T,g}} \mathcal{E}(h,t|\mathcal{S}) - \widehat{\mathcal{E}}(h,t|S) - \sup_{(h,t)\in\mathcal{H}^{T,g}} \mathcal{E}(h,t|\mathcal{S}) - \widehat{\mathcal{E}}(h,t|S')\Big|.$$

$$\leq \Big|\sup_{(h,t)\in\mathcal{H}^{T,g}} \Big(\mathcal{E}(h,t|\mathcal{S}) - \widehat{\mathcal{E}}(h,t|S) - \mathcal{E}(h,t|\mathcal{S}) + \widehat{\mathcal{E}}(h,t|S')\Big)\Big|.$$

$$= \Big|\sup_{(h,t)\in\mathcal{H}^{T,g}} \Big(\widehat{\mathcal{E}}(h,t|S) - \widehat{\mathcal{E}}(h,t|S')\Big)\Big|.$$

$$= \Big|\sup_{(h,t)\in\mathcal{H}^{T,g}} \Big(\frac{1}{|S|}\sum_{\mathbf{z}_j\in S}\ell_{0-1}^{\perp}(h,t,\mathbf{x}_j,y_j) - \frac{1}{|S'|}\sum_{\mathbf{z}_j\in S'}\ell_{0-1}^{\perp}(h,t,\mathbf{x}_j,y_j)\Big)\Big|.$$

$$= \left| \sup_{(h,t)\in\mathcal{H}^{T,g}} \left( \frac{1}{n}\sum_{j\neq i} \left(\ell^{\perp}_{0-1}(h,t,\mathbf{x}_j,y_j) - \ell^{\perp}_{0-1}(h,t,\mathbf{x}_j,y_j)\right) \right. \right. + $$

$$\left. \left. \frac{1}{n}\left(\ell^{\perp}_{0-1}(h,t,\mathbf{x}_i,y_i) - \ell^{\perp}_{0-1}(h,t,\mathbf{x}'_i,y'_i)\right) \right) \right|.$$

$$= \left| \sup_{(h,t)\in\mathcal{H}^{T,g}} \left( \frac{1}{n}\left(\ell^{\perp}_{0-1}(h,t,\mathbf{x}_i,y_i) - \ell^{\perp}_{0-1}(h,t,\mathbf{x}'_i,y'_i)\right) \right) \right|.$$

$$\leq \frac{1}{n}$$

The last step follows since $\ell^{\perp}_{0-1}$ is a 0-1 loss function so letting $\ell^{\perp}_{0-1}(h,t,\mathbf{x}_i,y_i) = 1$ and $\ell^{\perp}_{0-1}(h,t,\mathbf{x}'_i,y'_i) = 0$ gives an upper bound on the difference. Thus we can apply McDiarmid Inequality here and get the bound.

$\square$

The relationship between the Rademacher complexities is obtained using the following Lemma C.5 due to [16].

**Lemma C.5.** *([16]) Let $\ell_{0-1}, \ell_{\perp}, \ell^{\perp}_{0-1}$ be the loss functions defined as above and the Rademacher complexities on $n$ i.i.d. samples $S$ be $\mathfrak{R}_n(\mathcal{H},\ell_{0-1}), \mathfrak{R}_n(\mathcal{H}^{T,g},\ell_{\perp}), \mathfrak{R}_n(\mathcal{H}^{T,g},\ell^{\perp}_{0-1})$ respectively. Then,*

$$\mathfrak{R}_n(\mathcal{H}^{T,g},\ell^{\perp}_{0-1}) \leq \mathfrak{R}_n(\mathcal{H},\ell_{0-1}) + \mathfrak{R}_n(\mathcal{H}^{T,g},\ell_{\perp}) =: \mathfrak{R}_n(\mathcal{H}^{T,g}). \tag{13}$$

Detailed proof of this lemma can be found in [16]. The result follows by expressing $\ell_{0-1}\cdot\ell_{\perp}$ as $(\ell_{0-1} + \ell_{\perp} - 1)_+$ and then applying Talagrand's contraction lemma [39].

## C.2 Bounds for Finite VC-Dimension Classes

Here we specialize the auto-labeling error and coverage bounds to the setting of finite VC-dimension classes and then instantiate for a specific setting of homogeneous linear classifiers and uniform distribution.

**Lemma C.6.** *[43] (Corollary 3.8 and 3.18). Let the VC-dimension of function class induced by $\mathcal{F}$ be any class of functions from $\mathcal{X} \mapsto \mathcal{Y}\cup\{\perp\}$, and $\ell : \mathcal{Y}\cup\{\perp\} \mapsto \{0,1\}$ be a 0-1 function. Then,*

$$\mathfrak{R}_n(\mathcal{F},\ell) \leq \sqrt{\frac{2\mathcal{V}(\mathcal{F},\ell)}{n}\log\left(\frac{en}{\mathcal{V}(\mathcal{F},\ell)}\right)}. \tag{14}$$

**Corollary C.7.** *(Auto-Labeling Error and Coverage for Finite VC-dimension Classes) Let $k$ denote the number of rounds of TBAL algorithm 1. Let $\mathcal{V}(\mathcal{H}^{T,g}) = d$ Let $X_{pool}(A_k)$ be the set of auto-labeled points at the end of round $k$. $N_a^{(k)} = \sum_{i=1}^k n_a^{(i)}$ denote the total number of auto-labeled points. With probability at least $1 - \delta$,*

$$\widehat{\mathcal{E}}(X_{pool}(A_k)) \leq \sum_{i=1}^k \frac{n_a^{(i)}}{N_a^{(k)}} \left( \underbrace{\widehat{\mathcal{E}}_a(\hat{h}_i,\hat{t}_i|X_v^{(i)})}_{(a)} + \underbrace{\frac{4}{p_0}\sqrt{\frac{2}{n_v^{(i)}}\left(2d\log\left(\frac{en_v^{(i)}}{d}\right) + \log\left(\frac{8k}{\delta}\right)\right)}}_{(b)} \right)$$

$$+ \underbrace{\frac{4}{p_0}\left(\sqrt{\frac{2k}{N_a^{(k)}}\left(2d\log\left(\frac{eN_a^{(k)}}{d}\right) + \log\left(\frac{8k}{\delta}\right)\right)}\right)}_{c}$$

*and*

$$\widehat{\mathcal{P}}(X_{pool}(A_k)) \geq \sum_{i=1}^k \mathbb{P}\left(\mathcal{X}^{(i)}(\hat{h}_i,\hat{t}_i)\right) - 2k\sqrt{\frac{2}{N}\left(2d\log\left(\frac{eN}{d}\right) + \log\left(\frac{8k}{\delta}\right)\right)}.$$

*Proof.* The proof follows by substituting the Rademacher complexity bounds for finite VC dimension function classes from Lemma C.6 in the general result from Theorem 3.2.

$$\widehat{\mathcal{E}}(X_{pool}(A_k)) \leq \sum_{i=1}^{k} \frac{n_a^{(i)}}{N_a^{(k)}} \Bigg( \underbrace{\widehat{\mathcal{E}}_a(\hat{h}_i, \hat{t}_i | X_v^{(i)})}_{(a)} + \frac{4}{p_0} \big( \underbrace{\mathfrak{R}_{n_v^{(i)}}(\mathcal{H}^{T,g}) + \mathfrak{R}_{n_a^{(i)}}(\mathcal{H}^{T,g})}_{(b)} \big)$$

$$+ \underbrace{\frac{4}{p_0} \sqrt{\frac{1}{n_v^{(i)}} \log\left(\frac{8k}{\delta}\right)}}_{(c)} \Bigg) + \underbrace{\frac{4}{p_0} \sqrt{\frac{k}{N_a^{(k)}} \log\left(\frac{8k}{\delta}\right)}}_{(d)}$$

We first simplify the terms dependent on $n_v^{(i)}$ as follows. Here we use the inequality $\sqrt{a} + \sqrt{b} \leq \sqrt{2(a+b)}$ for any $a, b \in \mathbb{R}^+$.

$$\mathfrak{R}_{n_v^{(i)}}(\mathcal{H}^{T,g}) + \sqrt{\frac{1}{n_v^{(i)}} \log\left(\frac{8k}{\delta}\right)} \leq \sqrt{\frac{2d}{n_v^{(i)}} \log\left(\frac{en_v^{(i)}}{d}\right)} + \sqrt{\frac{1}{n_v^{(i)}} \log\left(\frac{4k}{\delta}\right)},$$

$$\leq \sqrt{\frac{2}{n_v^{(i)}} \left( 2d \log\left(\frac{en_v^{(i)}}{d}\right) + \log\left(\frac{8k}{\delta}\right) \right)}.$$

Next, we simplify the terms dependent on $n_a^{(i)}$ as follows. First, we substitute the Rademacher complexity using the bound in Lemma C.6 and then apply the same steps as in the proof of Theorem 3.2 to bound $\sum_{i=1}^{k} \sqrt{n_a^{(i)}/N_a^{(k)}}$ by $\sqrt{k/N_a^{(k)}}$ followed by the application of $\sqrt{a} + \sqrt{b} \leq \sqrt{2(a+b)}$ to get the final term.

$$\sum_{i=1}^{k} \frac{n_a^{(i)}}{N_a^{(k)}} \mathfrak{R}_{n_a^{(i)}}(\mathcal{H}^{T,g}) + \sqrt{\frac{k}{N_a^{(k)}} \log(\frac{8k}{\delta})} \leq \sum_{i=1}^{k} \frac{n_a^{(i)}}{N_a^{(k)}} \sqrt{\frac{2d}{n_a^{(i)}} \log\left(\frac{en_a^{(i)}}{d}\right)} + \sqrt{\frac{k}{N_a^{(k)}} \log\left(\frac{8k}{\delta}\right)}$$

$$= \sum_{i=1}^{k} \frac{\sqrt{n_a^{(i)}}}{N_a^{(k)}} \sqrt{2d \log\left(\frac{en_a^{(i)}}{d}\right)} + \sqrt{\frac{k}{N_a^{(k)}} \log\left(\frac{8k}{\delta}\right)}$$

$$\leq \sum_{i=1}^{k} \frac{\sqrt{n_a^{(i)}}}{N_a^{(k)}} \sqrt{2d \log\left(\frac{eN_a^{(k)}}{d}\right)} + \sqrt{\frac{k}{N_a^{(k)}} \log\left(\frac{8k}{\delta}\right)}$$

$$\leq \sqrt{\frac{2dk}{N_a^{(k)}} \log\left(\frac{eN_a^{(k)}}{d}\right)} + \sqrt{\frac{k}{N_a^{(k)}} \log\left(\frac{8k}{\delta}\right)}$$

$$\leq \sqrt{\frac{2k}{N_a^{(k)}} \left( 2d \log\left(\frac{eN_a^{(k)}}{d}\right) + \log\left(\frac{8k}{\delta}\right) \right)}.$$

$\square$

### C.3 Homogeneous Linear Classifiers with Uniform Distribution

Here we instantiate Theorem 3.2 for the case of homogeneous linear separators under the uniform distribution in the realizable setting. Formally, let $P_{\mathbf{x}}$ be a uniform distribution supported on the unit ball in $\mathbb{R}^d$, $\mathcal{X} = \{\mathbf{x} \in \mathbb{R}^d : ||\mathbf{x}|| \leq 1\}$. Let $\mathcal{W} = \{\mathbf{w} \in \mathbb{R}^d : ||\mathbf{w}||_2 = 1\} = \mathbb{S}_d$ and $\mathcal{H} = \{\mathbf{x} \mapsto \text{sign}(\langle \mathbf{w}, \mathbf{x} \rangle) \forall \mathbf{w} \in \mathcal{W}\}$, the score function is given by $g(h, \mathbf{x}) = g(\mathbf{w}, \mathbf{x}) = |\langle \mathbf{w}, \mathbf{x} \rangle|$ and set $T = [0, 1]$. For simplicity, we will use $\mathcal{W}$ in place of $\mathcal{H}$.

**Corollary 3.4.** *(Overall Auto-Labeling Error and Coverage) Let $\hat{\mathbf{w}}_i, \hat{t}_i$ be the ERM solution and the auto-labeling margin threshold respectively at epoch $i$. Let $n_v^{(i)}, n_a^{(i)}$ denote the number of validation and auto-labeled points at epoch $i$. Let the auto-labeling algorithm run for $k$-epochs. Then, for any*

$\delta \in (0,1)$, *w.p. at least* $1 - \delta/2$,

$$\widehat{\mathcal{E}}(X_{pool}(A_k)) \leq \sum_{i=1}^{k} \frac{n_a^{(i)}}{N_a^{(k)}} \left( \underbrace{\widehat{\mathcal{E}}_a(\hat{\mathbf{w}}_i, \hat{t}_i | X_v^{(i)})}_{(a)} + \underbrace{\frac{4}{p_0} \sqrt{\frac{2}{n_v^{(i)}} \left( 2d \log \left( \frac{en_v^{(i)}}{d} \right) + \log \left( \frac{8k}{\delta} \right) \right)}}_{(b)} \right)$$

$$+ \underbrace{\frac{4}{p_0} \left( \sqrt{\frac{2k}{N_a^{(k)}} \left( 2d \log \left( \frac{eN_a^{(k)}}{d} \right) + \log \left( \frac{8k}{\delta} \right) \right)} \right)}_{c}$$

*and w.p. at least* $1 - \delta/2$

$$\widehat{\mathcal{P}}(X_{pool}(A_k)) \geq 1 - \min_i \hat{t}_i \sqrt{4d/\pi} - 2k \sqrt{\frac{2}{N} \left( 2d \log \left( \frac{eN}{d} \right) + \log \left( \frac{8k}{\delta} \right) \right)}.$$

*Proof.* The bound on auto-labeling error follows directly from Theorem C.7 as the VC dimension for this setting is $d$. For the coverage bound, we utilize the fact that the distribution $P_{\mathbf{x}}$ is the uniform distribution over the unit ball. This enables us to obtain explicit lower bounds on the coverage. The details are given in Lemma C.8 and Lemma C.9. $\square$

**Lemma C.8.** *Let the auto-labeling algorithm run for $k$-epochs and let $\hat{\mathbf{w}}_i, \hat{t}_i$ be the ERM solution and the auto-labeling margin threshold respectively at epoch $i$. Let $\mathcal{X}^{(i)}$ be the unlabeled region at the beginning of epoch $i$, then we have,*

$$\sum_{i=1}^{k} \mathbb{P}\big(\mathcal{X}^{(i)}(\hat{\mathbf{w}}_i, \hat{t}_i)\big) \geq 1 - \min_i \hat{t}_i \sqrt{4d/\pi}. \tag{15}$$

*Proof.* Let $\mathcal{X}(\hat{\mathbf{w}}_i, t_i) = \{\mathbf{x} \in \mathcal{X} : |\langle \hat{\mathbf{w}}_i, \mathbf{x} \rangle| \geq \hat{t}_i\}$ denote the region that can be auto-labeled by $\hat{\mathbf{w}}_i, \hat{t}_i$. However, since in each round the remaining region is $\mathcal{X}^{(i)}$ the actual auto-labeled region of epoch $i$ is $\mathcal{X}_a^{(i)} = \{\mathbf{x} \in \mathcal{X}^{(i)} : |\langle \hat{\mathbf{w}}_i, \mathbf{x} \rangle| \geq \hat{t}_i\}$. Let $\bar{\mathcal{X}}(\hat{\mathbf{w}}_i, t_i)$ denote the complement of set $\mathcal{X}(\hat{\mathbf{w}}_i, t_i)$.

Now observe that $\mathcal{X}_a = \cup_{i=1}^{k} \mathcal{X}_a^{(i)}$ and $\mathcal{X}(\hat{\mathbf{w}}_k, \hat{t}_k) \subseteq \mathcal{X}_a$ because any $\mathbf{x} \in \mathcal{X}(\hat{\mathbf{w}}_k, \hat{t}_k)$ is either auto-labeled in previous rounds $i < k$ or if not then it will be auto-labeled in the $k^{th}$ round. More specifically, any $\mathbf{x} \in \mathcal{X}(\hat{\mathbf{w}}_k, \hat{t}_k)$ is either in $\cup_{i=1}^{k-1} \mathcal{X}_a^{(i)}$ and if not then it must be in $\mathcal{X}_a^{(k)}$. Thus the sum of probabilities,

$$\sum_{i=1}^{k} \mathbb{P}\big(\mathcal{X}^{(i)}(\hat{\mathbf{w}}_i, \hat{t}_i)\big) = \sum_{i=1}^{k} \mathbb{P}(\mathcal{X}_a^{(i)})$$
$$= \mathbb{P}(\mathcal{X}_a)$$
$$\geq \min_i \mathbb{P}\big(\mathcal{X}(\hat{\mathbf{w}}_i, \hat{t}_i)\big)$$
$$= 1 - \max_i \mathbb{P}\big(\bar{\mathcal{X}}(\hat{\mathbf{w}}_i, \hat{t}_i)\big)$$
$$\geq 1 - \min_i \hat{t}_i \sqrt{4d/\pi}$$

The last step used Lemma 4 from [3]) with $\gamma_1 = \hat{t}_i$ and $\gamma_2 = 0$ to upper bound $\mathbb{P}(\bar{\mathcal{X}}(\hat{\mathbf{w}}_i, \hat{t}_i))$ by $\hat{t}_i \sqrt{4d/\pi}$. The lemma is stated as follows in Lemma C.9, $\square$

**Lemma C.9.** *([3] (Lemma 4)) Let $d \geq 2$ and let $\mathbf{x} = [x_1, \ldots x_d]$ be uniformly distributed in the $d$-dimensional unit ball. Given $\gamma_1 \in [0,1], \gamma_2 \in [0,1]$, we have:*

$$\mathbb{P}\big((x_1, x_2) \in [0, \gamma_1] \times [\gamma_2, 1]\big) \leq \frac{\gamma_1 \sqrt{d}}{2\sqrt{\pi}} \exp \left( - \frac{(d-2)\gamma_2^2}{2} \right)$$

## C.4   Lower Bound

**Lemma 3.3.** *Let $c_1, c_2$ and $\sigma > 0$. Let $\mathbf{x}_i \in X$ be a set of $n$ i.i.d. points from $\mathcal{X}$ with corresponding true labels $y_i$. Given $(h, t) \in \mathcal{H}^{T,g}$, let $\mathbb{E}\big[\big(\ell_{0-1}^{\perp}(h, t, \mathbf{x}_i, y_i) - \mathcal{E}(h, t|\mathcal{X})\big)^2\big] = \sigma_i^2 > \sigma^2$ for every $\mathbf{x}_i$ for $\sigma_i > 0$ and let $\sum_i^n \sigma_i^2 \geq c_1$ then for every $\epsilon \in [0, \frac{\sum_{i=1}^n \sigma_i^2}{\sqrt{c_1}}]$ with $n_v < \frac{12\sigma^2}{\epsilon^2} \log(4c_2)$ the following holds w.p. at least $1/4$, $\mathcal{E}_a(h, t|\mathcal{X}) > \widehat{\mathcal{E}}_a(h, t|X) + \epsilon$.*

*Proof.*  It follows by application of Feller's result stated in lemma C.10. □

**Lemma C.10.** *(Feller, Lower Bound on Tail Probability of Sum of Independent Random Variables) There exists positive universal constants $c_1$ and $c_2$ such that for any set of independent random variables $X_1, \ldots, X_m$ satisfying $E[X_i] = 0$ and $|X_i| \leq M$, for every $i \in \{1, \ldots, m\}$, if $\sum_{i=1}^m \mathbb{E}[(X)_i^2] \geq c_1$, then for every $\epsilon \in [0, \frac{\sum_{i=1}^m \mathbb{E}[(X_i)^2]\}}{M\sqrt{c_1}}]$*

$$\mathbb{P}(\sum_{i=1}^m X_i > \epsilon) \geq c_2 \exp\left(\frac{-\epsilon^2}{12 \sum_{i=1}^m \mathbb{E}[(X_i)^2]}\right). \tag{16}$$

# D  Additional Experiments

In this section, we discuss additional experiments on the role of hypothesis class in auto-labeling datasets and experiments for studying the role of confidence function in auto-labeling. Finally, we visualize PaCMAP embeddings of the CIFAR-10 and MNIST data points to get a sense of auto-labeling regions in various rounds of the algorithm.

## D.1  Additional Experiments on Role of the Hypothesis Class

First, we provide details of the datasets,

**XOR** is a synthetic dataset. Recall that it is created by uniformly drawing points from 4 circles, each centered at the corners of a square of with side length 4 centered at the origin. Points in the diagonally opposite balls belong to the same class. We generate a total of $N = 10,000$ samples, out of which we keep $8,000$ in $X_{pool}$ and $2,000$ in the validation pool $X_{val}$.

**MNIST** [15] is a standard image dataset of hand-written digits. We randomly split the standard training set into $X_{pool}$ and the validation pool $X_{val}$ of sizes 48,000 and 12,000 respectively. While training a linear classifier on this dataset we flatten the $28 \times 28$ images to vectors of size 784.

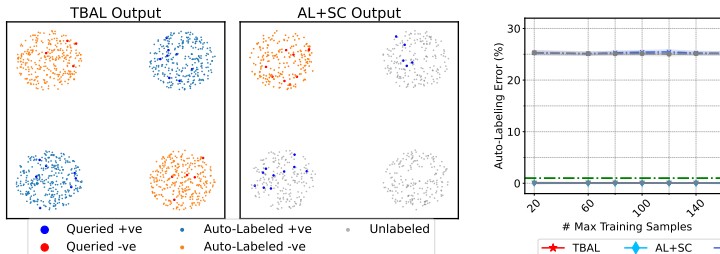

(a) Output of TBAL and AL+SC on XOR dataset        (b) Auto labeling performance of various methods

Figure 5: Comparison of Threshold-Based Auto-Labeling (TBAL) and Active-Learning followed by Selective Classification (AL+SC) on XOR-dataset. Left figure (a) shows samples that were auto-labeled, queried, and left unlabeled by these methods. Right figure (b) shows the auto-labeling error and coverage achieved. The lines show the mean and the shaded region shows 1-standard deviation estimated over 10 trials with different random seeds.

**XOR Experiment.** We run the TBAL algorithm 1 with an error tolerance of $\epsilon_a = 1\%$. we use 20% of $N_q$ as seed training data and keep query size $n_b$ as 5% of $N_q$. We compare it with active learning and active learning followed by selective classification. The given function class and selective classifier are both linear for all the algorithms. The results are shown in Figure 5. Clearly, there is no linear classifier that can correctly classify this data. We note that there are multiple optimal classifiers in the function class of linear classifiers and they will all incur an error of 25%. So, active learning algorithms can only output models that make at least 25% error. If we naively use the output model for auto-labeling, we can obtain near full coverage but incur 25% auto-labeling error. If we use the model output by active learning with threshold-based selective classification, then it can attain lower error in labeling. However, it can only label $\approx 25\%$ of the unlabeled data. In contrast, the TBAL algorithm can label almost all of the data accurately, i.e., attain close to $100\%$ coverage, with an error close to $1\%$ auto-labeling error.

**MNIST Experiment.** For training LeNet [37] we use SGD with a learning rate of 0.1, batch size of 32, and train for 20 epochs. We use auto-labeling error threshold $\epsilon_a = 5\%$. We use 20% of $N_q$ as seed training data and keep query size $n_b$ as 5% of $N_q$. The results are presented in Figure 6 we observe that TBAL using less powerful models can still yield highly accurate datasets with a significant fraction of points labeled by the models. This confirms the notion that bad models can still provide good datasets.

## D.2  Role of Confidence Function

The confidence function $g$ is used to obtain uncertainty scores is an important factor in auto-labeling. In particular, for threshold-based auto-labeling we expect the scores of correctly classified and incorrectly classified points to be reasonably well separated and if this is not the case then the

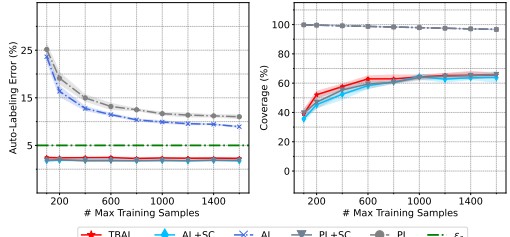 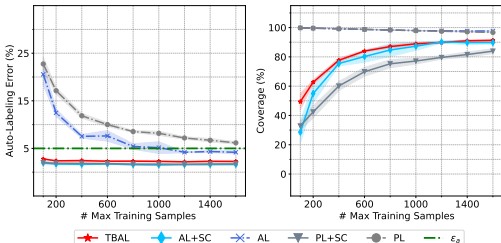

(a) Auto-labeling MNIST data using a linear classifier. The validation size used here is 12k.

(b) Auto-labeling MNIST data using LeNet classifier. The validation size used here is 12k.

Figure 6: Auto-labeling performance on MNIST data using different models (hypothesis classes) as a function of samples available for training. The left figure (a) shows the results with the linear classifier and the right figure (b) shows the results with the LeNet classifier. The auto-labeling error threshold $\epsilon_a = 5\%$ in both experiments and the algorithms are given the same amount of validation data. The lines show the mean and the shaded region shows 1-standard deviation estimated over 5 trials with different random seeds.

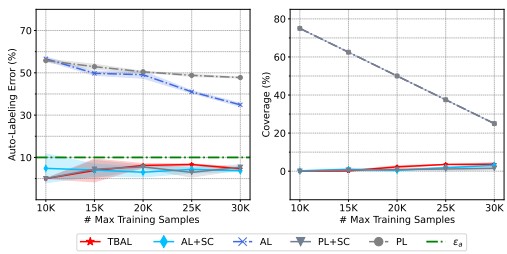 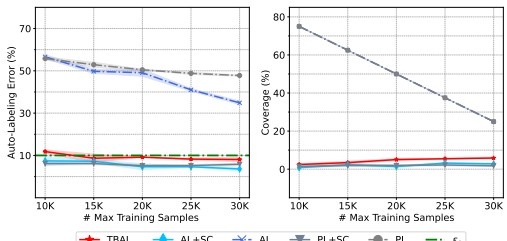

(a) Auto-labeling CIFAR-10 data using a small network and softmax scores. Validation size = 10k.

(b) Auto-labeling CIFAR-10 data using a small network and energy scores. Validation size = 10k.

Figure 7: Auto-labeling performance on CIFAR-10 data using a small network and different scoring functions. The left figure (a) shows the results with softmax scores and the right figure (b) shows the results with the energy score. The auto-labeling error threshold $\epsilon_a = 10\%$ in both experiments and methods are given the same amount of validation data. The lines show the mean and the shaded region shows 1-standard deviation estimated over 5 trials with different random seeds.

algorithm will struggle to find a good threshold even if the given classifier has good accuracy in certain regions.

**Setup.** We perform auto-labeling on the CIFAR-10 dataset using a small CNN network with 2 convolution layers followed by 3 fully connected layers [48]. We use two different scores for auto-labeling, a) Usual softmax output b) Energy score with temperature = 1 [38]. We vary the maximum number of training samples $N_q$ and keep 20% of $N_q$ as seed samples and query points in the batches of 10% of $N_q$. The model is trained for 50 epochs, using SGD with a learning rate of 0.05, batch size = 256, weight decay = $5e^{-4}$, and momentum=0.9, and use $\epsilon_a = 10\%$.

**Results.** The results with softmax scores and energy scores used as confidence functions can be seen in Figures 8(a) and 8(b) respectively. We see that for both of these cases, TBAL does not obtain a coverage of more than $\approx 6\%$. We observe that using the energy score as the confidence function performed marginally better than using the softmax scores. We note that this is the case even though the test accuracies of the trained models were around 50% for most of the rounds. Note that CIFAR-10 has 10 classes, so an accuracy of 50% is much better than random guessing and one would expect to be able to auto-label a significant chunk of the data with such a model. However, the softmax scores and energy scores are not well calibrated, and therefore, when used as confidence functions, they result in a poor separation between correct and incorrect predictions by the model. This can be seen in Figure 8 where neither of the softmax and energy scores provides a good separation between the correct and incorrect predictions. We can also see that the energy score is marginally better in terms of the separation, which allows it to achieve slightly better auto-labeling coverage in comparison to using softmax scores. This suggests that more investigation is needed to understand the properties of good confidence functions for auto-labeling which is left to future work.

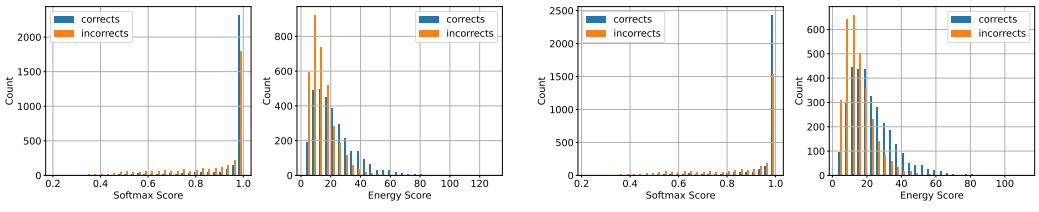

(a) Histogram of scores in round 2.    (b) Histogram of scores in round 6.

Figure 8: Histograms of scores computed on the validation data in a few rounds of TBAL run on CIFAR-10 with a small net. We picked two rounds where it auto-labeled the most i.e. around 800 points.

For a more detailed visualization of the rounds of TBAL for this experiment, see Figures 10 and 11 in the Appendix.

### D.3 Detailed Results

In the main paper, we omitted AL, PL, and PL+SC for clarity and due to lack of space. We provide results including these baselines in the following tables for IMDB, Tiny-ImageNet, and Unit-Ball datasets.

| $N_v$ | Error (%) | | | | | Coverage (%) | | | | |
|---|---|---|---|---|---|---|---|---|---|---|
| | TBAL | AL+SC | PL+SC | AL | PL | TBAL | AL+SC | PL+SC | AL | PL |
| 200 | 4.77 $_{\pm0.18}$ | 3.35 $_{\pm0.80}$ | 3.68 $_{\pm0.38}$ | 6.60 $_{\pm1.06}$ | 6.16 $_{\pm0.10}$ | 83.14 $_{\pm3.65}$ | 78.53 $_{\pm7.05}$ | 81.20 $_{\pm1.78}$ | 98.75 $_{\pm0.00}$ | 97.50 $_{\pm0.00}$ |
| 400 | 4.57 $_{\pm0.26}$ | 3.53 $_{\pm0.73}$ | 3.96 $_{\pm0.33}$ | 6.60 $_{\pm1.06}$ | 6.16 $_{\pm0.10}$ | 90.70 $_{\pm3.11}$ | 86.39 $_{\pm5.11}$ | 87.43 $_{\pm2.01}$ | 98.75 $_{\pm0.00}$ | 97.50 $_{\pm0.00}$ |
| 600 | 4.32 $_{\pm0.17}$ | 3.70 $_{\pm0.63}$ | 4.17 $_{\pm0.27}$ | 6.60 $_{\pm1.06}$ | 6.16 $_{\pm0.10}$ | 92.96 $_{\pm0.46}$ | 88.90 $_{\pm4.83}$ | 90.12 $_{\pm1.62}$ | 98.75 $_{\pm0.00}$ | 97.50 $_{\pm0.00}$ |
| 800 | 4.66 $_{\pm0.20}$ | 3.84 $_{\pm0.70}$ | 4.15 $_{\pm0.35}$ | 6.60 $_{\pm1.06}$ | 6.16 $_{\pm0.10}$ | 92.42 $_{\pm0.89}$ | 88.67 $_{\pm3.88}$ | 89.37 $_{\pm1.45}$ | 98.75 $_{\pm0.00}$ | 97.50 $_{\pm0.00}$ |
| 1000 | 4.67 $_{\pm0.16}$ | 3.90 $_{\pm0.68}$ | 4.21 $_{\pm0.35}$ | 6.60 $_{\pm1.06}$ | 6.16 $_{\pm0.10}$ | 92.89 $_{\pm0.91}$ | 89.79 $_{\pm3.09}$ | 90.18 $_{\pm1.39}$ | 98.75 $_{\pm0.00}$ | 97.50 $_{\pm0.00}$ |

Table 6: **IMDB**. Effect of variation of validation data size ($N_v$) without using a UCB (i.e., $C_1 = 0$) on error estimates. We keep training data size $N_q$ fixed at 500 and use error threshold $\epsilon_a = 5\%$. We report the mean and std. deviation over 10 runs with different random seeds.

| $N_v$ | Error (%) | | | | | Coverage (%) | | | | |
|---|---|---|---|---|---|---|---|---|---|---|
| | TBAL | AL+SC | PL+SC | AL | PL | TBAL | AL+SC | PL+SC | AL | PL |
| 200 | 2.28 $_{\pm0.21}$ | 3.11 $_{\pm0.86}$ | 2.86 $_{\pm0.35}$ | 6.60 $_{\pm1.06}$ | 6.16 $_{\pm0.10}$ | 68.24 $_{\pm6.20}$ | 57.77 $_{\pm13.09}$ | 60.45 $_{\pm1.63}$ | 98.75 $_{\pm0.00}$ | 97.50 $_{\pm0.00}$ |
| 400 | 1.29 $_{\pm0.10}$ | 1.98 $_{\pm0.40}$ | 1.54 $_{\pm0.11}$ | 6.60 $_{\pm1.06}$ | 6.16 $_{\pm0.10}$ | 63.81 $_{\pm4.86}$ | 63.06 $_{\pm10.70}$ | 68.32 $_{\pm6.60}$ | 98.75 $_{\pm0.00}$ | 97.50 $_{\pm0.00}$ |
| 600 | 1.41 $_{\pm0.20}$ | 1.81 $_{\pm0.22}$ | 1.87 $_{\pm0.07}$ | 6.60 $_{\pm1.06}$ | 6.16 $_{\pm0.10}$ | 69.64 $_{\pm3.98}$ | 62.92 $_{\pm9.20}$ | 69.84 $_{\pm3.07}$ | 98.75 $_{\pm0.00}$ | 97.50 $_{\pm0.00}$ |
| 800 | 1.62 $_{\pm0.30}$ | 2.04 $_{\pm0.35}$ | 2.33 $_{\pm0.35}$ | 6.60 $_{\pm1.06}$ | 6.16 $_{\pm0.10}$ | 67.45 $_{\pm3.72}$ | 63.22 $_{\pm7.89}$ | 72.50 $_{\pm2.28}$ | 98.75 $_{\pm0.00}$ | 97.50 $_{\pm0.00}$ |
| 1000 | 1.64 $_{\pm0.23}$ | 1.97 $_{\pm0.26}$ | 1.93 $_{\pm0.13}$ | 6.60 $_{\pm1.06}$ | 6.16 $_{\pm0.10}$ | 70.28 $_{\pm2.82}$ | 66.11 $_{\pm8.00}$ | 73.04 $_{\pm2.06}$ | 98.75 $_{\pm0.00}$ | 97.50 $_{\pm0.00}$ |

Table 7: **IMDB**. Effect of variation of validation data size ($N_v$), using a UCB (i.e., $C_1 = 0.25$) on error estimates. We keep training data size $N_q$ fixed at 500 and use error threshold $\epsilon_a = 5\%$. We report the mean and std. deviation over 10 runs with different random seeds.

| $N_q$ | Error (%) | | | | | Coverage (%) | | | | |
|---|---|---|---|---|---|---|---|---|---|---|
| | TBAL | AL+SC | PL+SC | AL | PL | TBAL | AL+SC | PL+SC | AL | PL |
| 200 | 4.57 $_{\pm0.21}$ | 4.30 $_{\pm0.20}$ | 3.97 $_{\pm0.06}$ | 6.44 $_{\pm0.20}$ | 6.20 $_{\pm0.09}$ | 93.46 $_{\pm1.01}$ | 90.60 $_{\pm2.91}$ | 91.97 $_{\pm0.28}$ | 99.50 $_{\pm0.00}$ | 99.00 $_{\pm0.00}$ |
| 400 | 4.60 $_{\pm0.09}$ | 3.75 $_{\pm0.83}$ | 3.92 $_{\pm0.86}$ | 11.86 $_{\pm13.54}$ | 6.81 $_{\pm1.26}$ | 92.55 $_{\pm0.66}$ | 84.27 $_{\pm19.15}$ | 89.95 $_{\pm2.92}$ | 99.00 $_{\pm0.00}$ | 98.00 $_{\pm0.00}$ |
| 600 | 4.93 $_{\pm0.10}$ | 3.99 $_{\pm0.91}$ | 4.69 $_{\pm0.09}$ | 6.31 $_{\pm1.28}$ | 6.33 $_{\pm0.10}$ | 92.45 $_{\pm0.84}$ | 91.69 $_{\pm3.99}$ | 91.20 $_{\pm0.42}$ | 98.50 $_{\pm0.00}$ | 97.00 $_{\pm0.00}$ |
| 800 | 4.76 $_{\pm0.12}$ | 3.55 $_{\pm0.69}$ | 4.37 $_{\pm0.14}$ | 6.91 $_{\pm1.49}$ | 6.12 $_{\pm0.10}$ | 92.15 $_{\pm1.05}$ | 89.98 $_{\pm3.38}$ | 89.97 $_{\pm0.69}$ | 98.00 $_{\pm0.00}$ | 96.00 $_{\pm0.00}$ |
| 1000 | 4.49 $_{\pm0.06}$ | 4.19 $_{\pm0.31}$ | 4.25 $_{\pm0.29}$ | 5.65 $_{\pm0.25}$ | 6.14 $_{\pm0.11}$ | 92.25 $_{\pm0.96}$ | 92.28 $_{\pm2.13}$ | 89.47 $_{\pm0.70}$ | 97.50 $_{\pm0.00}$ | 95.00 $_{\pm0.00}$ |

Table 8: **IMDB**. Effect of variation of $N_q$, the maximum number of samples the algorithm can use for training, without using a UCB (i.e., $C_1 = 0$) on error estimates. We keep validation data size $N_v$ fixed at 1000 and use error threshold $\epsilon_a = 5\%$. We report the mean and std. deviation over 10 runs with different random seeds.

| $N_q$ | Error (%) | | | | | Coverage (%) | | | | |
|---|---|---|---|---|---|---|---|---|---|---|
| | TBAL | AL+SC | PL+SC | AL | PL | TBAL | AL+SC | PL+SC | AL | PL |
| 200 | 1.67 $_{\pm0.29}$ | 2.15 $_{\pm0.45}$ | 1.59 $_{\pm0.10}$ | 6.44 $_{\pm0.20}$ | 6.20 $_{\pm0.09}$ | 73.30 $_{\pm3.49}$ | 57.17 $_{\pm11.09}$ | 57.39 $_{\pm4.15}$ | 99.50 $_{\pm0.00}$ | 99.00 $_{\pm0.00}$ |
| 400 | 1.63 $_{\pm0.19}$ | 1.61 $_{\pm0.29}$ | 1.76 $_{\pm0.13}$ | 11.86 $_{\pm13.54}$ | 6.81 $_{\pm1.26}$ | 72.59 $_{\pm3.16}$ | 64.53 $_{\pm16.61}$ | 58.48 $_{\pm1.79}$ | 99.00 $_{\pm0.00}$ | 98.00 $_{\pm0.00}$ |
| 600 | 1.67 $_{\pm0.21}$ | 1.83 $_{\pm0.30}$ | 1.67 $_{\pm0.08}$ | 6.31 $_{\pm1.28}$ | 6.33 $_{\pm0.10}$ | 71.38 $_{\pm2.15}$ | 70.50 $_{\pm5.68}$ | 65.71 $_{\pm2.14}$ | 98.50 $_{\pm0.00}$ | 97.00 $_{\pm0.00}$ |
| 800 | 1.67 $_{\pm0.27}$ | 1.90 $_{\pm0.31}$ | 1.79 $_{\pm0.09}$ | 6.91 $_{\pm1.49}$ | 6.12 $_{\pm0.10}$ | 69.10 $_{\pm4.51}$ | 65.74 $_{\pm10.14}$ | 73.21 $_{\pm2.57}$ | 98.00 $_{\pm0.00}$ | 96.00 $_{\pm0.00}$ |
| 1000 | 1.62 $_{\pm0.22}$ | 1.97 $_{\pm0.35}$ | 1.70 $_{\pm0.12}$ | 5.65 $_{\pm0.25}$ | 6.14 $_{\pm0.11}$ | 73.42 $_{\pm2.84}$ | 68.05 $_{\pm5.56}$ | 64.18 $_{\pm2.11}$ | 97.50 $_{\pm0.00}$ | 95.00 $_{\pm0.00}$ |

Table 9: **IMDB**. Effect of variation of $N_q$, the maximum number of samples the algorithm can use for training, using a UCB (i.e., $C_1 = 0.25$) on error estimates. We keep validation data size $N_v$ fixed at 1000 and use error threshold $\epsilon_a = 5\%$. We report the mean and std. deviation over 10 runs with different random seeds.

### D.4  Auto Labeling Visualization

In this section, we visualize the process of TBAL. We use the dimensionality reduction method, PaCMAP [66], to visualize the features of the samples. For neural network models, we visualize the PaCMAP embeddings of the penultimate layer's output and for linear models, we use PaCMAP on the raw features. In these figures, each row corresponds to one TBAL round. Each figure shows a few selected rounds of auto-labeling. Each figure has four columns (left to right), which show: **a**) The samples that are labeled by TBAL in the round are shown in that row. **b**) The embeddings for training samples in that round. **c**) The embeddings for validation data points in that round. **d**) The score distribution for the validation dataset in that round.

In Figure 9 we see visualizations for auto-labeling on the MNIST data using linear models. In this setting the data exhibits clustering structure in the PaCMAP embeddings learned on the raw features and the confidence (probability) scores produced are also reasonably well calibrated which leads to good auto-labeling performance.

The visualizations for the process of TBAL on CIFAR-10 using the small network (a small CNN network with 2 convolution layers followed by 3 fully connected layers [48]) with energy scores and soft-max scores for confidence functions are shown in Figures 10 and 11 respectively. We note that both the energy scores and soft-max scores do not seem to be calibrated to the correctness of the predicted labels which makes it difficult to identify subsets of unlabeled data where the current hypothesis in each round could have potentially auto-labeled. We also note that the test accuracies of the trained models were around $50\%$ for most of the rounds of TBAL even though the small network model is not a powerful enough model class for this dataset. Note that CIFAR-10 has 10 classes, so the accuracy of $50\%$ is much better than random guessing and one would expect to be able to auto-label a reasonably large chunk of the data with such a model if accompanied by a good confidence function. This highlights the important role that the confidence function plays in a TBAL system and more investigation is needed which is left to future work.

Note that, in our auto-labeling implementation we find class specific thresholds. In these figures, we show the histograms of scores for all classes for simplicity. We want to emphasize that the visualization figures in this section are 2D representations (approximation) of the high-dimensional features (either of the penultimate layer or the raw features).

| $N_v$ | Error (%) | | | | | Coverage (%) | | | | |
|---|---|---|---|---|---|---|---|---|---|---|
| | TBAL | AL+SC | PL+SC | AL | PL | TBAL | AL+SC | PL+SC | AL | PL |
| 100 | 3.10 ±1.80 | 0.68 ±0.81 | 1.45 ±0.73 | 1.23 ±0.99 | 2.87 ±0.57 | 71.43 ±8.86 | 96.95 ±1.01 | 92.29 ±3.27 | 98.52 ±0.16 | 96.88 ±0.00 |
| 400 | 1.97 ±0.76 | 0.59 ±0.18 | 1.19 ±0.53 | 0.81 ±0.26 | 2.87 ±0.57 | 93.99 ±2.39 | 97.89 ±0.50 | 91.73 ±2.86 | 98.44 ±0.00 | 96.88 ±0.00 |
| 800 | 1.64 ±0.50 | 0.66 ±0.19 | 1.21 ±0.41 | 0.81 ±0.26 | 2.87 ±0.57 | 96.26 ±1.33 | 98.06 ±0.53 | 92.25 ±2.31 | 98.44 ±0.00 | 96.88 ±0.00 |
| 1200 | 1.39 ±0.39 | 0.67 ±0.19 | 1.11 ±0.30 | 0.81 ±0.26 | 2.87 ±0.57 | 96.67 ±0.84 | 98.10 ±0.45 | 91.98 ±2.20 | 98.44 ±0.00 | 96.88 ±0.00 |
| 1600 | 1.33 ±0.30 | 0.70 ±0.19 | 1.11 ±0.26 | 0.81 ±0.26 | 2.87 ±0.57 | 97.13 ±0.45 | 98.16 ±0.44 | 92.01 ±2.09 | 98.44 ±0.00 | 96.88 ±0.00 |
| 2000 | 1.28 ±0.34 | 0.71 ±0.21 | 1.07 ±0.25 | 0.81 ±0.26 | 2.87 ±0.57 | 97.15 ±0.54 | 98.20 ±0.34 | 91.86 ±2.17 | 98.44 ±0.00 | 96.88 ±0.00 |

Table 10: **Unit Ball**. Effect of variation of validation data size ($N_v$), without using a UCB (i.e., $C_1 = 0$) on error estimates. We keep training data size $N_q$ fixed at 500 and use error threshold $\epsilon_a = 1\%$. We report the mean and std. deviation over 10 runs with different random seeds.

| $N_v$ | Error (%) | | | | | Coverage (%) | | | | |
|---|---|---|---|---|---|---|---|---|---|---|
| | TBAL | AL+SC | PL+SC | AL | PL | TBAL | AL+SC | PL+SC | AL | PL |
| 100 | 3.10 ±1.80 | 0.68 ±0.81 | 1.45 ±0.73 | 1.23 ±0.99 | 2.87 ±0.57 | 71.43 ±8.86 | 96.95 ±1.01 | 92.29 ±3.27 | 98.52 ±0.16 | 96.88 ±0.00 |
| 400 | 1.65 ±0.65 | 0.32 ±0.15 | 0.52 ±0.32 | 0.81 ±0.26 | 2.87 ±0.57 | 93.27 ±2.50 | 96.91 ±0.99 | 87.86 ±3.73 | 98.44 ±0.00 | 96.88 ±0.00 |
| 800 | 1.08 ±0.47 | 0.24 ±0.16 | 0.31 ±0.17 | 0.81 ±0.26 | 2.87 ±0.57 | 96.01 ±1.16 | 96.31 ±1.36 | 86.21 ±3.55 | 98.44 ±0.00 | 96.88 ±0.00 |
| 1200 | 0.78 ±0.27 | 0.17 ±0.11 | 0.18 ±0.14 | 0.81 ±0.26 | 2.87 ±0.57 | 96.82 ±0.84 | 95.96 ±1.40 | 84.65 ±4.14 | 98.44 ±0.00 | 96.88 ±0.00 |
| 1600 | 0.65 ±0.20 | 0.13 ±0.08 | 0.12 ±0.09 | 0.81 ±0.26 | 2.87 ±0.57 | 96.93 ±0.57 | 95.70 ±1.38 | 83.76 ±3.93 | 98.44 ±0.00 | 96.88 ±0.00 |
| 2000 | 0.54 ±0.16 | 0.21 ±0.11 | 0.21 ±0.10 | 0.81 ±0.26 | 2.87 ±0.57 | 97.23 ±0.42 | 96.36 ±1.13 | 85.72 ±3.47 | 98.44 ±0.00 | 96.88 ±0.00 |

Table 11: **Unit-Ball**. Effect of variation of validation data size ($N_v$), without using a UCB (i.e., $C_1 = 0.25$) on error estimates. We keep training data size $N_q$ fixed at 500 and use error threshold $\epsilon_a = 1\%$. We report the mean and std. deviation over 10 runs with different random seeds.

| $N_q$ | Error (%) | | | | | Coverage (%) | | | | |
|---|---|---|---|---|---|---|---|---|---|---|
| | TBAL | AL+SC | PL+SC | AL | PL | TBAL | AL+SC | PL+SC | AL | PL |
| 100 | 1.53 ±0.27 | 1.16 ±0.35 | 1.14 ±0.29 | 16.93 ±2.48 | 12.53 ±2.24 | 75.31 ±7.06 | 31.69 ±10.78 | 51.47 ±10.46 | 99.69 ±0.00 | 99.38 ±0.00 |
| 200 | 1.25 ±0.21 | 1.04 ±0.25 | 0.98 ±0.17 | 7.85 ±1.69 | 7.08 ±1.82 | 96.24 ±0.88 | 73.27 ±8.68 | 76.05 ±8.19 | 99.38 ±0.00 | 98.75 ±0.00 |
| 400 | 1.20 ±0.20 | 0.94 ±0.17 | 1.03 ±0.15 | 1.81 ±0.58 | 3.25 ±0.81 | 97.70 ±0.19 | 96.48 ±1.74 | 91.08 ±2.63 | 98.75 ±0.00 | 97.50 ±0.00 |
| 600 | 1.21 ±0.26 | 0.44 ±0.13 | 1.09 ±0.18 | 0.44 ±0.14 | 2.25 ±0.56 | 97.56 ±0.20 | 98.11 ±0.02 | 93.19 ±1.60 | 98.12 ±0.00 | 96.25 ±0.00 |
| 800 | 1.13 ±0.20 | 0.12 ±0.05 | 1.02 ±0.18 | 0.12 ±0.05 | 1.76 ±0.50 | 97.25 ±0.19 | 97.49 ±0.01 | 93.23 ±1.01 | 97.50 ±0.00 | 95.00 ±0.00 |
| 1000 | 1.08 ±0.19 | 0.04 ±0.02 | 1.00 ±0.20 | 0.04 ±0.02 | 1.37 ±0.34 | 97.02 ±0.25 | 96.87 ±0.01 | 92.90 ±0.79 | 96.88 ±0.00 | 93.75 ±0.00 |

Table 12: **Unit-Ball**. Effect of variation of $N_q$, the maximum number of samples the algorithm can use for training, without using a UCB (i.e., $C_1 = 0$) on error estimates. We keep validation data size $N_v$ fixed at 4000 and use error threshold $\epsilon_a = 1\%$. We report the mean and std. deviation over 10 runs with different random seeds.

| $N_q$ | Error (%) | | | | | Coverage (%) | | | | |
|---|---|---|---|---|---|---|---|---|---|---|
| | TBAL | AL+SC | PL+SC | AL | PL | TBAL | AL+SC | PL+SC | AL | PL |
| 100 | 0.78 ±0.19 | 0.29 ±0.25 | 0.24 ±0.15 | 16.93 ±2.48 | 12.53 ±2.24 | 71.40 ±5.64 | 19.31 ±8.77 | 35.90 ±10.66 | 99.69 ±0.00 | 99.38 ±0.00 |
| 200 | 0.49 ±0.13 | 0.16 ±0.09 | 0.17 ±0.08 | 7.85 ±1.69 | 7.08 ±1.82 | 95.28 ±1.29 | 59.41 ±10.32 | 63.77 ±9.15 | 99.38 ±0.00 | 98.75 ±0.00 |
| 400 | 0.40 ±0.12 | 0.15 ±0.07 | 0.15 ±0.08 | 1.81 ±0.58 | 3.25 ±0.81 | 97.63 ±0.24 | 92.06 ±2.63 | 83.64 ±4.76 | 98.75 ±0.00 | 97.50 ±0.00 |
| 600 | 0.34 ±0.13 | 0.12 ±0.05 | 0.17 ±0.08 | 0.44 ±0.14 | 2.25 ±0.56 | 97.36 ±0.20 | 97.07 ±0.57 | 87.92 ±2.68 | 98.12 ±0.00 | 96.25 ±0.00 |
| 800 | 0.28 ±0.10 | 0.07 ±0.03 | 0.15 ±0.06 | 0.12 ±0.05 | 1.76 ±0.50 | 97.10 ±0.23 | 97.36 ±0.11 | 88.96 ±2.14 | 97.50 ±0.00 | 95.00 ±0.00 |
| 1000 | 0.25 ±0.10 | 0.03 ±0.02 | 0.19 ±0.08 | 0.04 ±0.02 | 1.37 ±0.34 | 96.90 ±0.21 | 96.84 ±0.03 | 89.61 ±1.49 | 96.88 ±0.00 | 93.75 ±0.00 |

Table 13: **Unit-Ball**. Effect of variation of $N_q$, the maximum number of samples the algorithm can use for training, using a UCB (i.e., $C_1 = 0.25$) on error estimates. We keep validation data size $N_v$ fixed at 4000 and use error threshold $\epsilon_a = 1\%$. We report the mean and std. deviation over 10 runs with different random seeds.

| $N_v$ | Error (%) | | | | | Coverage (%) | | | | |
|---|---|---|---|---|---|---|---|---|---|---|
| | **TBAL** | **AL+SC** | **PL+SC** | **AL** | **PL** | **TBAL** | **AL+SC** | **PL+SC** | **AL** | **PL** |
| 2000 | 0.0 ±0.0 | 0.0 ±0.0 | 0.0 ±0.0 | 0.0 ±0.0 | 0.0 ±0.0 | 0.0 ±0.0 | 0.0 ±0.0 | 0.0 ±0.0 | 0.0 ±0.0 | 0.0 ±0.0 |
| 4000 | 13.88 ±5.42 | 13.31 ±10.79 | 12.55 ±7.23 | 34.34 ±0.32 | 31.46 ±0.20 | 0.72 ±0.55 | 0.48 ±0.04 | 0.69 ±0.39 | 95.00 ±0.00 | 90.00 ±0.00 |
| 6000 | 14.18 ±0.76 | 11.52 ±0.82 | 12.21 ±1.49 | 34.42 ±0.34 | 31.46 ±0.20 | 17.29 ±0.72 | 8.18 ±1.12 | 11.81 ±2.70 | 95.00 ±0.00 | 90.00 ±0.00 |
| 8000 | 13.97 ±0.14 | 11.31 ±0.51 | 12.22 ±0.61 | 34.42 ±0.34 | 31.46 ±0.20 | 36.36 ±1.78 | 23.40 ±1.15 | 29.99 ±0.97 | 95.00 ±0.00 | 90.00 ±0.00 |
| 10000 | 13.42 ±0.29 | 11.14 ±0.54 | 12.12 ±0.40 | 34.42 ±0.34 | 31.46 ±0.20 | 43.79 ±0.93 | 33.38 ±0.72 | 39.40 ±0.50 | 95.00 ±0.00 | 90.00 ±0.00 |

Table 14: **Tiny-ImageNet**. Effect of variation of validation data size ($N_v$), without using a UCB (i.e., $C_1 = 0$) on error estimates. We keep training data size $N_q$ fixed at $10K$ and use error threshold $\epsilon_a = 10\%$. We report the mean and std. deviation over 10 runs with different random seeds.

| $N_v$ | Error (%) | | | | | Coverage (%) | | | | |
|---|---|---|---|---|---|---|---|---|---|---|
| | **TBAL** | **AL+SC** | **PL+SC** | **AL** | **PL** | **TBAL** | **AL+SC** | **PL+SC** | **AL** | **PL** |
| 2000 | 0.0 ±0.0 | 0.0 ±0.0 | 0.0 ±0.0 | 0.0 ±0.0 | 0.0 ±0.0 | 0.0 ±0.0 | 0.0 ±0.0 | 0.0 ±0.0 | 0.0 ±0.0 | 0.0 ±0.0 |
| 4000 | 10.50 ±6.01 | 7.37 ±4.57 | 6.04 ±1.85 | 34.20 ±0.32 | 31.52 ±0.27 | 0.47 ±0.05 | 0.48 ±0.06 | 0.43 ±0.01 | 95.00 ±0.00 | 90.00 ±0.00 |
| 6000 | 10.61 ±0.62 | 7.71 ±1.03 | 8.53 ±1.70 | 34.42 ±0.34 | 31.46 ±0.20 | 10.16 ±1.10 | 4.31 ±1.10 | 7.13 ±1.18 | 95.00 ±0.00 | 90.00 ±0.00 |
| 8000 | 9.90 ±0.63 | 6.80 ±0.77 | 7.81 ±0.85 | 34.42 ±0.34 | 31.46 ±0.20 | 25.84 ±1.57 | 14.43 ±2.01 | 19.23 ±1.43 | 95.00 ±0.00 | 90.00 ±0.00 |
| 10000 | 8.97 ±0.36 | 6.87 ±0.48 | 7.32 ±0.49 | 34.42 ±0.34 | 31.46 ±0.20 | 32.19 ±1.34 | 21.96 ±1.35 | 27.01 ±0.98 | 95.00 ±0.00 | 90.00 ±0.00 |

Table 15: **Tiny-ImageNet**. Effect of variation of validation data size ($N_v$), using a UCB (i.e., $C_1 = 0.25$) on error estimates. We keep training data size $N_q$ fixed at $10K$ and use error threshold $\epsilon_a = 10\%$. We report the mean and std. deviation over 10 runs with different random seeds.

| $N_q$ | Error (%) | | | | | Coverage (%) | | | | |
|---|---|---|---|---|---|---|---|---|---|---|
| | **TBAL** | **AL+SC** | **PL+SC** | **AL** | **PL** | **TBAL** | **AL+SC** | **PL+SC** | **AL** | **PL** |
| 2000 | 14.02 ±0.26 | 11.49 ±0.80 | 12.17 ±0.35 | 52.34 ±1.16 | 42.94 ±0.26 | 24.34 ±0.86 | 14.41 ±1.00 | 25.13 ±0.58 | 99.00 ±0.00 | 98.00 ±0.00 |
| 4000 | 14.10 ±0.77 | 11.58 ±0.26 | 11.92 ±0.39 | 43.14 ±0.33 | 36.07 ±0.41 | 34.16 ±1.00 | 21.84 ±1.36 | 33.41 ±0.65 | 98.00 ±0.00 | 96.00 ±0.00 |
| 6000 | 13.55 ±0.17 | 11.33 ±0.35 | 12.31 ±0.16 | 38.73 ±0.59 | 33.51 ±0.19 | 37.80 ±1.05 | 28.59 ±1.53 | 38.14 ±0.85 | 97.00 ±0.00 | 94.00 ±0.00 |
| 8000 | 13.79 ±0.27 | 11.72 ±0.32 | 12.36 ±0.30 | 36.06 ±0.30 | 32.33 ±0.32 | 42.00 ±1.71 | 32.00 ±1.12 | 39.64 ±1.07 | 96.00 ±0.00 | 92.00 ±0.00 |
| 10000 | 13.26 ±0.35 | 11.42 ±0.28 | 12.14 ±0.45 | 34.27 ±0.21 | 31.47 ±0.17 | 43.63 ±0.38 | 33.80 ±0.82 | 39.23 ±0.37 | 95.00 ±0.00 | 90.00 ±0.00 |

Table 16: **Tiny-ImageNet**. Effect of variation of $N_q$, the maximum number of samples the algorithm can use for training, without using a UCB (i.e., $C_1 = 0$) on error estimates. We keep validation data size $N_v$ fixed at $10K$ and use error threshold $\epsilon_a = 10\%$. We report the mean and std. deviation over 5 runs with different random seeds.

| $N_q$ | Error (%) | | | | | Coverage (%) | | | | |
|---|---|---|---|---|---|---|---|---|---|---|
| | **TBAL** | **AL+SC** | **PL+SC** | **AL** | **PL** | **TBAL** | **AL+SC** | **PL+SC** | **AL** | **PL** |
| 2000 | 9.22 ±1.04 | 7.42 ±0.71 | 7.48 ±0.32 | 52.34 ±1.16 | 42.94 ±0.26 | 17.51 ±1.16 | 9.33 ±0.66 | 17.02 ±1.32 | 99.00 ±0.00 | 98.00 ±0.00 |
| 4000 | 9.30 ±0.38 | 6.97 ±0.39 | 7.37 ±0.21 | 43.14 ±0.33 | 36.07 ±0.41 | 25.01 ±1.20 | 14.25 ±1.71 | 22.29 ±0.61 | 98.00 ±0.00 | 96.00 ±0.00 |
| 6000 | 9.12 ±0.22 | 6.85 ±0.26 | 7.49 ±0.35 | 38.73 ±0.59 | 33.51 ±0.19 | 28.06 ±0.75 | 17.51 ±0.36 | 25.60 ±0.34 | 97.00 ±0.00 | 94.00 ±0.00 |
| 8000 | 9.21 ±0.14 | 7.38 ±0.53 | 7.71 ±0.25 | 36.06 ±0.30 | 32.33 ±0.32 | 30.88 ±0.64 | 21.18 ±0.90 | 27.26 ±0.78 | 96.00 ±0.00 | 92.00 ±0.00 |
| 10000 | 8.95 ±0.23 | 7.10 ±0.26 | 7.42 ±0.36 | 34.27 ±0.21 | 31.47 ±0.17 | 32.31 ±1.21 | 22.34 ±0.61 | 27.36 ±0.59 | 95.00 ±0.00 | 90.00 ±0.00 |

Table 17: **Tiny-ImageNet**. Effect of variation of $N_q$, the maximum number of samples the algorithm can use for training, using a UCB (i.e., $C_1 = 0.25$) on error estimates. We keep validation data size $N_v$ fixed at $10K$ and use error threshold $\epsilon_a = 10\%$. We report the mean and std. deviation over 5 runs with different random seeds.

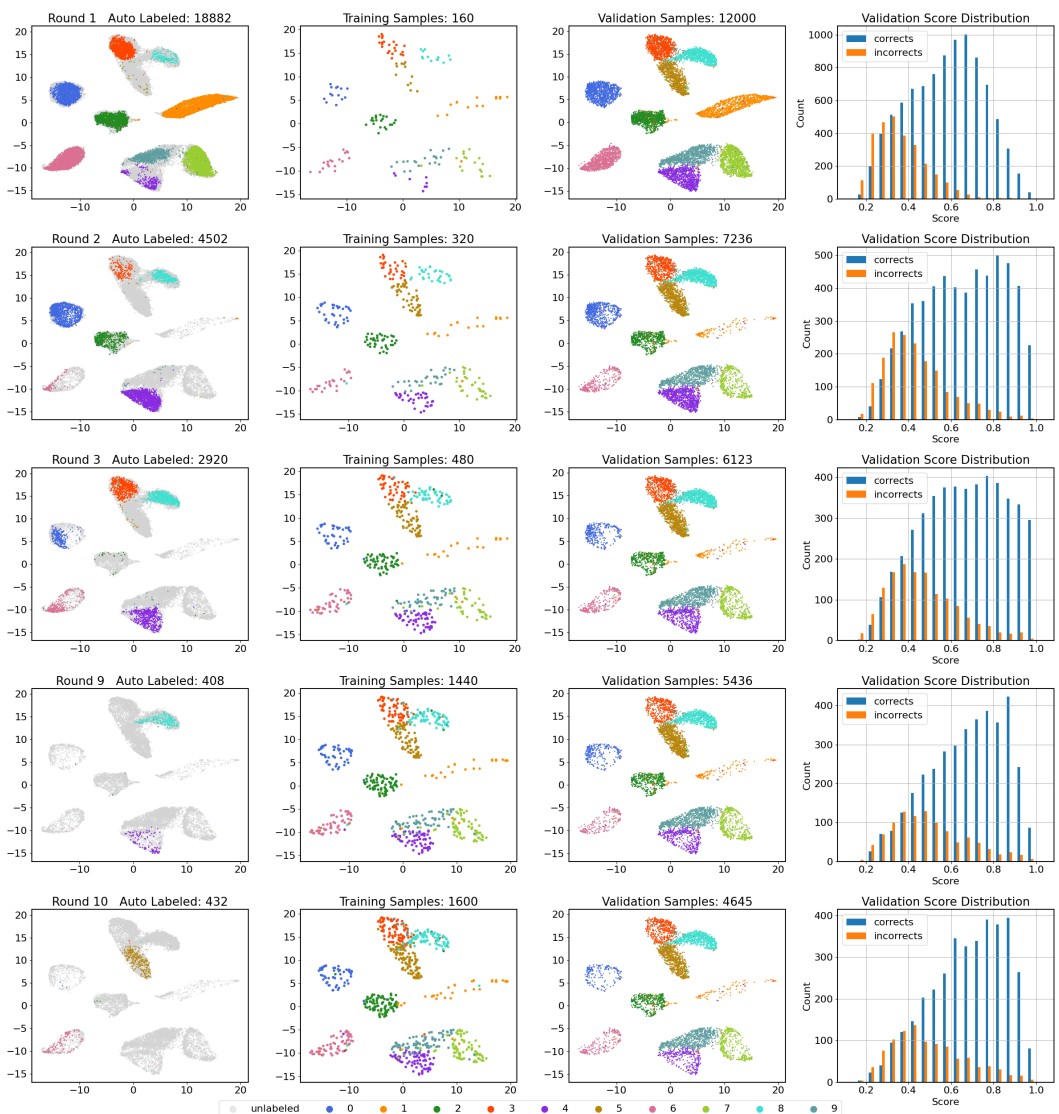

Figure 9: Auto-labeling MNIST data using linear classifiers. Validation size = 12k. Maximum training samples = 1600. Each round algorithm queries 160 samples. Coverage of auto-labeling is 62.9% with 98.0% accuracy. For the rounds we show, the test error rates are 21.4%, 13.9%, 12.5%, 10.2%, and 9.8%, respectively. For four columns (left to right), we show: **a**) The samples that are labeled by TBAL in this round. **b**) The embeddings for training samples. **c**) The embeddings for validation data points. **d**) The score distribution for the validation dataset.

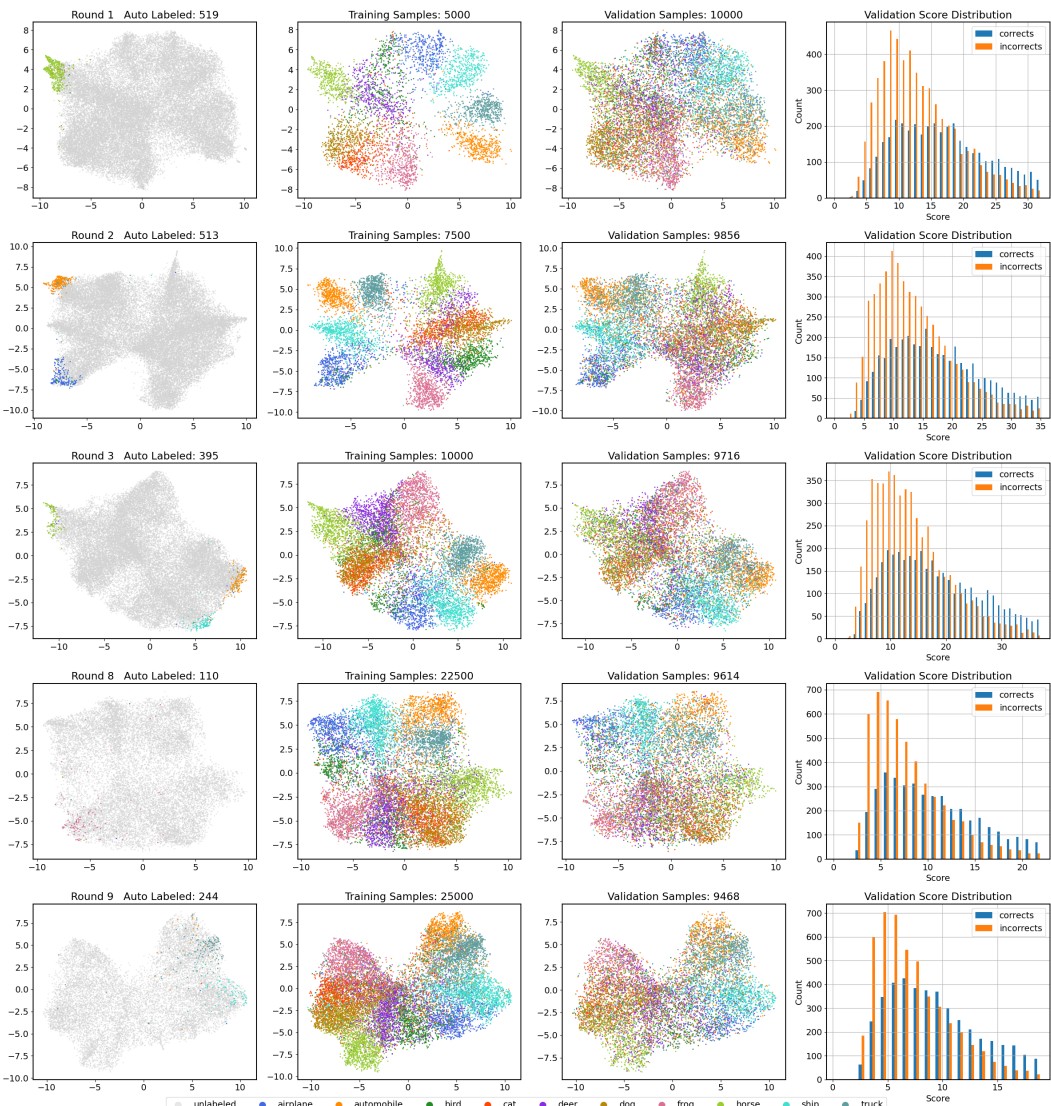

Figure 10: Auto-labeling CIFAR-10 data using a small network and energy scores. Validation size = 10k. Maximum training samples = 25k. Each round algorithm queries 2500 samples. Coverage of auto-labeling is 5.3% with 90.0% accuracy. For the rounds we show, the test error rates are 56.6%, 55.2%, 55.6%, 53.0%, and 49.3% respectively. For four columns (left to right), we show: **a**) The samples that are labeled by TBAL in this round. **b**) The embeddings for training samples. **c**) The embeddings for validation data points. **d**) The score distribution for the validation dataset.

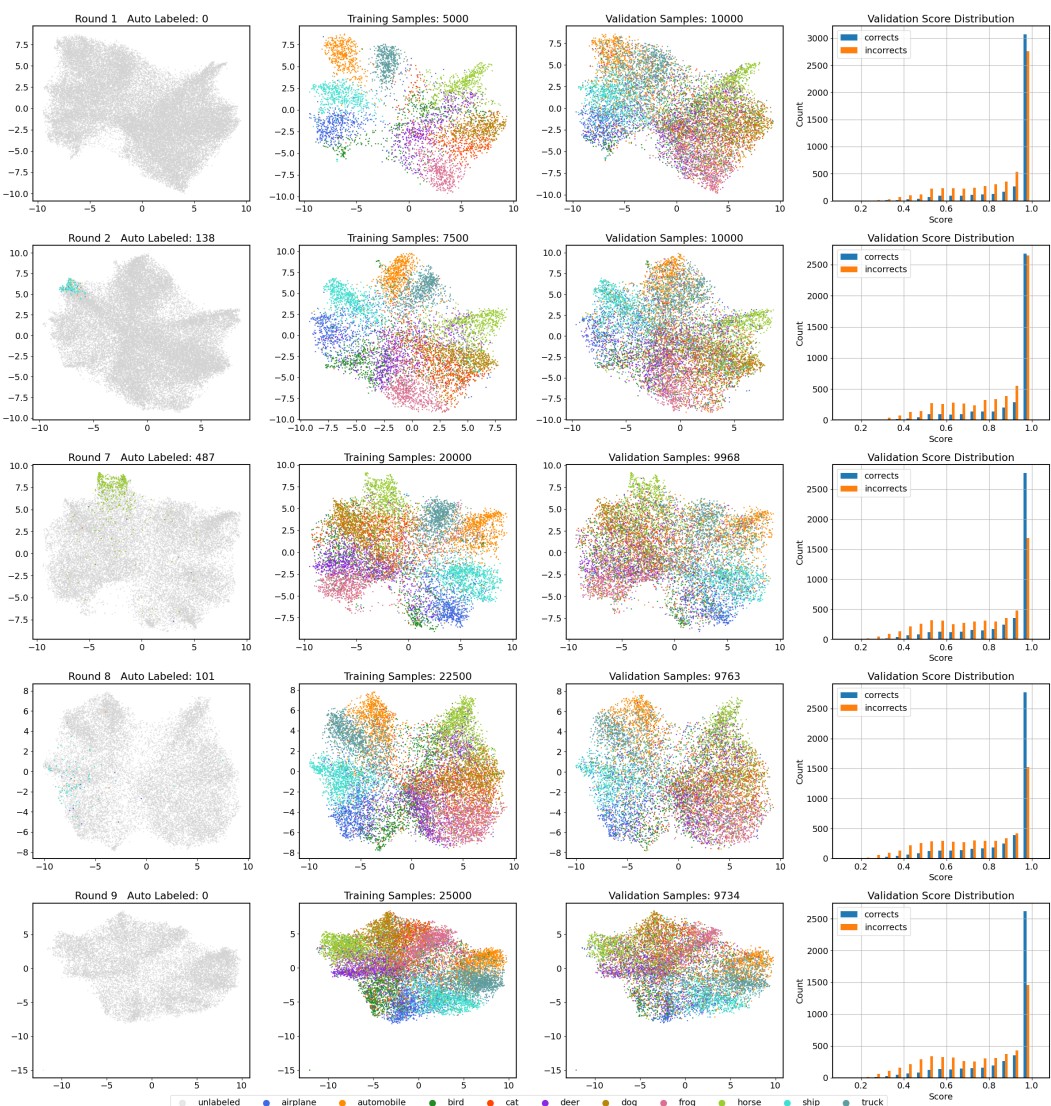

Figure 11: Auto-labeling CIFAR-10 data using a small network and softmax scores. Validation size = 10k. Maximum training samples = 25k. Each round algorithm queries 2500 samples. Coverage of auto-labeling is 2.3% with 91.0% accuracy. For the rounds visualized here in each row, the test error rates of the trained classifiers are 56.6%, 59.1%, 52.8%, 50.5%, and 51.7% respectively. For four columns (left to right), we show: **a**) The samples that are labeled by TBAL in this round. **b**) The embeddings for training samples. **c**) The embeddings for validation data points. **d**) The score distribution for the validation dataset.

