# OpenReview forum: "Promises and Pitfalls of Threshold-based Auto-labeling"
_NeurIPS.cc/2023/Conference — NeurIPS 2023 spotlight_

### Official Review · Reviewer_L2k7 · 2023-06-24

**Soundness:** 2 fair
**Presentation:** 3 good
**Contribution:** 3 good
**Rating:** 6
**Confidence:** 4

**Summary:**

The paper studies an important problem. It theoretically analyzes TBAL systems and develops tradeoffs between
the quantity of manually-labeled data and the quantity and quality of auto-labeled data. Empirical results on real/synthetic data also support their findings.
Their results provide two crucial insights. when having access to sufficient validation data and a good confidence function, even bad
models can reliably label at least some data, meanwhile, the quantity requirement of the validation data can be high.

**Strengths:**

1. This paper is the first to study the theoretic performance of TBAL, which is important in practice.
2. For the proposed TBAL, this paper gives theoretic guarantees on the auto-labeling error and the coverage using Rademacher Complexity and PAC theory.
3. The paper designs a series of experiments including synthetic data, and real data. The experiments do reveal some interesting observations (though some observations are unexpected, need more disscusions, see W3.)
4. The writing is good, easy to understand

**Weaknesses:**

W1. The goal of the paper seems a little confusing. On the one hand, it aims to study an existing popular TBAL system.
On the other hand, it seems that the studied TBAL algorithm is in fact proposed in this current paper.

W2. Several questions need to be clarified (possibly in the related work section):
(1) Is the proposed TBAL the same with any existing popular TBAL?
(2) Whether the theories apply to an existing popular TBAL system, what if their workflows are different?
(3) The paper claims that the TBAL is popular. But it didn't provide any evidence for its popularity, e.g., use cases, scenarios.

W3. In experiments, it seems AL+SC can outperform TBAL (in terms of error rate, given that the difference in coverage is not significant).
This is very different from Figure 2, why this can happen? Please add more discussions for explaining the different performance among the 4 competing methods.


**Questions:**

See weaknesses.

**Limitations:**

yes

---

> ### Author Rebuttal · Authors · 2023-08-10
>
> We sincerely appreciate your feedback and acknowledgment of the paper's strengths. Being the first theoretical study of TBAL systems, our research offers essential guarantees on auto-labeling error and coverage using learning theoretic tools. The experiments conducted not only validate our theoretical insights but also reveal intriguing observations regarding TBAL behavior. We are delighted that you found our writing to be clear and easily understandable.
>
> **Clarification on the goal of our work**: We would like to emphasize that the goal of our paper is to understand the pros and cons of the TBAL systems. **We are not proposing** this system.
>
> **Popularity of TBAL**: These systems are indeed used heavily in practice and this is what inspired us to study them. Since these are commercial systems, only high level descriptions of their methods are available. In our work we closely follow the workflow of Amazon Sagemaker GroundTruth [1] which is getting used in various domains including telecom, healthcare, recruiting etc (see their customers [2]). Similar workflows are also adopted in other systems e.g. Samsung SDS [3], Labelbox [4], V7[5] etc.
>
> **Comparison of AL+SC and TBAL**: Our goal is not to show TBAL is better than the other methods across the board. However, we are interested in understanding the behavior in TBAL in various settings and we compare it against AL+SC, which is a strong baseline for TBAL.
>
> We observe, in settings similar to Figure 2, where the model class does not contain a model with error at most $\epsilon$ then the end model in AL will have error more than $\epsilon$ and will lead to lesser coverage than TBAL. On the other hand the dynamics of TBAL are very different, it iteratively auto-labels and eliminates the data which enables it to achieve high coverage even in settings where the model class does not contain a model with error at most $\epsilon$. These differences can be easily seen Figure 2. However, in settings where the model class has models with error at most $\epsilon$ then AL+SC and TBAL will have similar performances. Please see table 1 and 2 in the attached pdf to see more clear comparison between these methods.
>
> [1] https://aws.amazon.com/blogs/machine-learning/annotate-data-for-less-with-amazon-sagemaker-ground-truth-and-automated-data-labeling/
>
> [2] https://aws.amazon.com/sagemaker/data-labeling/
>
> [3] https://www.samsungsds.com/us/autolabel/autolabel.html
>
> [4] https://labelbox.com/product/annotate/
>
> [5] https://www.v7labs.com/auto-annotation
>
>
> We appreciate your positive feedback and insightful questions, which have enhanced our paper. We are excited to answer any additional questions during the discussion phase.

---

### Official Review · Reviewer_zavU · 2023-07-05

**Soundness:** 3 good
**Presentation:** 3 good
**Contribution:** 3 good
**Rating:** 7
**Confidence:** 3

**Summary:**

This paper provides an analysis of Threshold-Based Auto-Labeling (TBAL) systems by deriving sample complexity bounds on the amount of human-labeled validation data required for guaranteeing the quality of the machine-labeled data. The paper provides two main insights: (1) large amounts of unlabeled data can be automatically/accurately labeled by weak models, and (2) a downside of TBAL systems is the need of potentially prohibitive amounts of necessary validation data.

**Strengths:**

To the best of this reviewer's knowledge, this is the first paper that derives sample complexity bounds on the amount of validation data required for the confident use of TBAL systems. The contributions appear to be novel, and the paper is well-written & reasonably easy to follow. The authors provide both a theoretical analysis and an empirical validation for TBAL systems. Figure 1 is extremely intuitive and summarizes very well TBAL systems for readers unfamiliar with the problem.

**Weaknesses:**

The paper could further benefit from additional discussion along the key findings.

1) while Fig 2.a is crisp & intuitive, Fig 2.b is less so. The authors should (1) explain why the coverage of AL & PL is decreasing as the training sample increases, and (2) discuss why the coverage of AL+DC & PL+SC does not really change for various amounts of training data

2) for Fig 3, the authors should (1) drop PL+SC with a note saying that AL+SC is always comparable/better, (2) explain why the coverage is so low for both IMDB & Tiny ImageNet, (3) discuss the usefulness of a TBAL approach with a coverage of about 25%, and (4) "zoom in" on an in-depth, 1:1 comparison between TBAL and AL+SC, which seem to be performing quite similarly (TBAL with a slightly better coverage, while AL+SC with a slightly better auto-labeling error); this comparison & potential trade-offs analysis should represent a major contribution of the paper.



**Questions:**

1) on line 268, the paper says that it is using five datasets, but it only lists four (Unit-Ball, Tiny-Imagenet, IMDB Reviews, and CIFAR-10). In the APPENDICES, the authors also bring up XOR & MNIST, for a total of six, rather than five, datasets

2) In the main paper, why don't you show the results for all 4 datasets (see above)?

3) would a scatter plot of AutoLabel Error vs Coverage help illuminate any trade-offs/insights for the domains in which the error/coverage are not totally/mostly flat?

4) line 261: could you please explain why is Algo 2 is estimating a single threshold for all classes, while in Section 4 you estimate per-class thresholds?



**Limitations:**

Could you please discuss in depth the practical impact/usefulness of a TBAL-created dataset that has only 25% coverage?

---

> ### Author Rebuttal · Authors · 2023-08-10
>
> We are delighted to receive your positive feedback and recognition of the strengths in our paper.
> Our work represents the first attempt to derive sample complexity bounds on the validation data for the confident utilization of TBAL systems. Moreover, we are happy to know that the paper's clarity and coherence have made it reasonably easy to follow. We are pleased that Figure 1 has been regarded as extremely intuitive and useful, particularly for readers who may not be familiar with the problem. It was designed with the intent to provide an accessible summary of TBAL systems and their workings, and we are glad it has served its purpose effectively.
> We appreciate your questions and suggestions and address them below,
>
> **On coverage of AL & PL:** This is due to the way we compute coverage – with more points taken into the training pool there are lesser points available for auto-labeling. This leads to a decrease in coverage as all the training samples available for that run are used. We have added discussion on this in the paper to further clarify.
>
> **On coverage of AL+SC & PL+SC:** There are two factors that lead to the observed behavior. First, due to the choice of the model class there is a limitation on the best model that can be learned irrespective of the training data provided i.e. the Bayes optimal classifier in this setting has error around 25% and AL, PL methods can only learn a classifier which is close to the Bayes optimal classifier. Once they reach close to the optimal classifier with more training data the improvements in the model will be negligible. Second, these methods only do a single round of auto-labeling i.e. learn the best possible model first and then do auto-labeling using this model. As we saw in the first point they hit a limit on the best model that can be learned, which leads to a limit on the coverage that they can achieve.
>
> **Suggested drop of PL+SC:** Thank you for the suggestion. We have updated the draft accordingly.
>
> **Why the coverage is low for both IMDB & Tiny ImageNet:** The coverage depends on several other factors, e.g. feature separability, confidence functions etc. In this work we wanted to study the role of validation data in the estimation of threshold estimation and overall auto-labeling performance.  We used a more expressive feature extractor in the IMDB experiment and saw a massive boost in the auto-labeling performance. Results can be seen in the attached pdf. This additional experiment serves as a further insight into the possibilities of TBAL systems.
>
> **On usefulness of TBAL with 25% coverage:** The coverage achieved by TBAL depends on many factors including the label noise tolerance, separability of features etc.
>
> In the context of data-labeling with crowd-sourcing, this fraction could mean a non-trivial reduction in the cost. 25% of what are large datasets are still quite sizable, implying that it may be possible to train a high-quality downstream classifier on a provably-clean dataset.  Even in cases where 25% coverage is too low for great downstream performance, our approach clarifies the tradeoffs between dataset size and label noise. That is, we could vary our confidence levels to achieve higher noise/higher coverage and see which is the "right" setting for a particular task and model. Such a setting may be more robust to noise but more data hungry, while others may be hurt by noise but robust to dataset size. Our approach disambiguates these---which wasn't possible before.
>
> **On an in-depth, 1:1 comparison between TBAL and AL+SC:** We show the data in tables in the attached pdf.
>
> **Number of datasets:**  Thanks for pointing this out. We have made these corrections in the updated draft.
>
> **On a scatter plot of AutoLabel Error vs Coverage:** Please see our new experiments in the attached pdf.
>
> **On why is Algo 2  estimating a single threshold for all classes:** We did this for easier exposition of the analysis.
>
> Once again, we sincerely thank you for their positive feedback and questions, which have further strengthened our paper. We are excited to engage in the discussion and answer any additional questions.

---

### Official Review · Reviewer_RfNU · 2023-07-06

**Soundness:** 4 excellent
**Presentation:** 3 good
**Contribution:** 3 good
**Rating:** 7
**Confidence:** 4

**Summary:**

Threshold-based auto-labelling (TBAL) is widely adopted in academic and industrial settings and this is the first work that studies TBAL from a theoretical standpoint, answering the questions of 1) Does TBAL works even if our models are not perfect and 2) what are the key requirements to having a successful TBAL pipeline? The paper suggests that even with a low-performant model, TBAL still helps accelerates the annotations. However, to facilitate a successful TBAL system, a potentially prohibitively large validation dataset might be needed. After anchoring the keys of TBAL, the paper experiments with several simple settings to validate the effectiveness of TBAL.


**Strengths:**

TBAL is widely adopted in both academic and industrial settings and the paper is the first that study it theoretically. The derivation is sound and can be easily drawn connections with the active learning literature. I expect more follow-up work in the future.
The strength of this work include
- Novelty: first work to study TBAL theoretically
- Clarity: The delivery is easy to follow

**Weaknesses:**

Although the paper is novel, section 3 and section 4 are disconnected to me. Section 3 gives us a bound on the error rate and coverage of TBAL, which provides insightful relations between the performance and different design factors.
However, I cannot find the bound in Figure 3 and Figure 4. I expect to see how tight the bound is empirically.

On the other hand, Eq. 2 also provides insights into the empirical validation error (term a). I expect the authors to push the limits for using a more expressive feature extractor and discuss the relation between the empirical validation error and the TBAL error rate empirically.


**Questions:**

- L204 suggests to use a hypothesis class that induces lower Rademacher complexities to reduce the second term in Eq. 2. However, a hypothesis class (e.g., Lipchitz-constrained neural networks) with lower Rademacher complexities usually leads to higher validation errors, therefore, increasing the first term in Eq. 1. If so, do the authors still suggest practitioners to use a hypothesis class that indices lower Rademacher complexities. How to balance the trade-off here?
    - This matters since a practitioner wants a rule of thumb for using TBAL. It’s important to discuss the trade-off here.
- Minor: As mentioned in #weakness, the current experimental results show that annotating 5k TinyImageNet effectively requires a validation dataset with 5k images, which does not sound “promising”.  Have the authors tried to push the limits a little bit further by using a better feature extractor?


**Limitations:**

The paper provides a novel view of TBAL, but the theoretical and empirical parts are not connected well. The authors should provide more numbers on how tight the proposed bound is under auto-labelling.

---

> ### Author Rebuttal · Authors · 2023-08-10
>
> Thank you for your valuable feedback and for recognizing the strengths of our paper. We greatly appreciate your acknowledgment of the significance of TBAL, which is heavily used in practice. As you rightly pointed out, our paper is the first theoretical study of TBAL and provides an understanding of conditions when we can ensure performance. We appreciate your comments on the soundness and clarity of the paper.
>
> **On tightness of the bounds:** Thank you for asking this question. We study this in the setting of the Unit-Ball experiment discussed in the main paper. The upper bound on excess risk in this setting is given in Corollary 3.4 which is an instantiation of our general results to this specific setting. We consider a simplified form of the upper bound by ignoring the constants to get a sense of the rate in terms of the validation data size. We compute this simplified upper bound for different amounts of validation data. We compare these with the maximum auto-labeling error observed over 25 runs of auto-labeling in the Unit-Ball setting with different random seeds for each validation data size. The results are in Figure 1 in the attached pdf.  As expected, we see that the worst-case error rate follows a similar rate as our upper bound but the upper bound is conservative. Next, we provide an explanation for why this is the case.
>
> Our upper bound is slightly conservative, as it is based on a uniform bound over all hypotheses in a given hypothesis class. Since the individual hypotheses whose excess auto-labeling error we need to bound are not known a priori we need to derive a bound on the number of validation samples using which we can guarantee that the excess auto-labeling errors of **any** hypothesis (model) is small with high probability. Note that this is a conservative (worst-case) analysis to get an upper bound on the validation sample complexity. The upper bound has two parts: a) the Rademacher complexity of the hypothesis class and b) a term with the number of validation samples.  We note that on the validation samples, it **matches lower bounds order-wise (see lemma 3.3)**.  This is the first analysis to provide these bounds based on uniform convergence **without making any assumptions on the data distributions or hypothesis class**. Tighter results are possible if we assume specific settings, and we shall pursue this in the future.
>
> **On using more expressive feature extractors:**
> Thank you for the valuable suggestion. We used a more expressive feature extractor in the IMDB experiment and saw a massive boost in the auto-labeling performance. Results can be seen in the attached pdf. We use the *bge-large-en* model from the MTEB leaderboard [1]. This additional experiment serves as a further insight into the possibilities of TBAL systems.
>
> [1] https://huggingface.co/spaces/mteb/leaderboard
>
> **Rademacher complexity and validation error trade-off:** Thank you for raising this interesting question. The bound contains validation errors at different thresholds. We revisit the definitions of validation errors appearing in the bound.  Let $X_v =$ { $x_1,\ldots x_{n_v}$} be the validation samples and $y_i$ be the label corresponding to $x_i$. Given any $h\in \mathcal{H}$ and the confidence function $g$, we have different subsets $X_v(h,t)$  of the validation points for which the model’s confidence is higher than $t$. More precisely, $X_v(h,t) =$ { $x_i \in X_v : g(h,x_i)\ge t$} and the validation error $\hat{\mathcal{E}}_a(h,t|X_v)$ is computed on each of these subsets as follows,
>
> $\hat{M}(h,t|X_v) = \sum_{x_i \in X_v(h,t)} \mathbf{1}( h(x_i) \neq y_i)$,
> $\quad{\hat{\mathcal{E}}}_{a}(h,t|X_v) = \frac{\hat{M}(h,t | X_v)}{|X_v(h,t)|}$.
>
> This error is different from the overall validation error which is computed over the entire set of validation points $X_v$:
>
> $\hat{\mathcal{E}}(h|X_v) = \frac{1}{|X_v|} \sum_{x_i \in X_v} \mathbf{1}(h(x_i)\neq y_i)$.
>
> The TBAL method computes $\hat{\mathcal{E}}_a(h,t|X_v)$ at different thresholds ($t$) and selects the threshold at which it is at most $\epsilon$. Thus even if the overall validation error $\hat{\mathcal{E}}(h|X_v)$ is bad, there could still be regions in the space where the conditional validation error $\hat{\mathcal{E}}_a(h,t|X_v) $ is small. This can be easily seen in Figure 2 in the paper. Here we are doing auto-labeling using a linear function class that has low Rademacher complexity. All the models in this class have high overall validation error  $\hat{\mathcal{E}}(h|X_v)$ but there are subsets where the conditional validation error  $\hat{\mathcal{E}}_a(h,t|X_v)$ is small and TBAL is able to find those subsets.
>
> Thus we see that working with low Rademacher complexity classes might lead to high overall validation error. However, it does not affect TBAL as long as there are regions of low conditional validation error $\hat{\mathcal{E}}_a(h,t|X_v) $. Furthermore, our upper bound on excess auto-labeling error depends on $\hat{\mathcal{E}}_a(h,t|X_v) $ which due to the TBAL procedure is at most $\epsilon$ and hence it is not in conflict with the Rademacher complexity term.
>
> Once again, we sincerely thank you for the positive feedback and questions. These insightful comments have helped us to further strengthen our paper. We are excited to engage in the discussion and answer further questions.

---

> > ### Comment · Reviewer_RfNU · 2023-08-18
> > **Review response**
> >
> > I am very satisfied with your reply. This is a good article and very enlightening! I have a habit of giving conservative marks in the first stage review! I hope it will not affect you badly.
> > I will raise my rating to 7 (Accept).

---

### Author Rebuttal · Authors · 2023-08-10

Thank you all for the thoughtful and positive feedback on our paper. With your constructive suggestions, we have been able to further strengthen our work with additional experiments and improved clarity.

Your questions and comments have provided us with valuable insights and we answer each of them in the individual responses. In this common response, we note the strengths that you have highlighted in your reviews and provide a summary of the individual responses.

**Strengths of our work as noted in your reviews:**


1. **Novelty**: Our paper is the first to study the widely used TBAL systems [RfNU, zavU, L2k7].


2. **Theoretical analysis of TBAL**: We derive sample complexity bounds on the amount of validation data required for the confident use of TBAL systems. In other words, we provide theoretical guarantees on the auto-labeling error and the coverage using Rademacher Complexity and PAC theory  [RfNU,zavU, L2k7].


3. **Empirical evaluation of TBAL**: Our paper provides an extensive set of experiments on synthetic and real datasets that evaluate TBAL in a variety of settings and validate our theoretical insights [zavU, L2k7]


4. **Clarity** :  All the reviewers found our paper well written and easy to follow.  [RfNU,zavU, L2k7]. We are delighted to hear that reviewer zavU found that “Figure 1 is extremely intuitive and summarizes very well TBAL systems for readers unfamiliar with the problem”.


In response to the reviewers' suggestions, we have done additional experiments and updated the draft to include those. Some of the experiments are included in the attached pdf file. These new results and clarifications have further strengthened our paper.

---

### Author Response · Authors · 2023-08-12
**Request to engage in discussion**

Dear Area Chair,

We would love it if you could remind reviewers to kick off conversations with us!

We believe we could answer any remaining questions and provide further clarification. Additional feedback from reviewers would be invaluable.

We would particularly love to engage with reviewers E4A7 and DxCE during the discussion period.

Thank you so much for your help!

The Authors

---

> ### Comment · Reviewer_zavU · 2023-08-17
> **Still ACCEPT**
>
> I would like to thank the authors for their careful rebuttals. Their comments and the reviews of my colleagues are reinforcing the ACCEPT status of this paper

---

### Decision · Program_Chairs · 2023-09-21

**Decision:**

Accept (spotlight)

**Comment:**

This paper on threshold-based autolabeling studies an important and understudied problem.
The reviewers are unanimous that the paper is interesting, clearly presented, and warrants publication.